# Simplicity Bias via Global Convergence of Sharpness Minimization

## Abstract

The remarkable generalization ability of neural networks is usually attributed to the implicit bias of SGD, which often yields models with lower complexity using simpler (e.g. linear) and low-rank features Huh et al. (2021). Recent works have provided empirical and theoretical evidence for the bias of particular variants of SGD (such as label noise SGD) toward flatter regions of the loss landscape. Despite the folklore intuition that flat solutions are 'simple', the connection with the simplicity of the final trained model (e.g. low-rank) is not well understood. In this work, we take a step toward bridging this gap by studying the *simplicity structure* that arises from minimizers of the sharpness for a class of two-layer neural networks. We show that, for any high dimensional training data and certain activations, with small enough step size, label noise SGD always converges to a network that replicates a single linear feature across all neurons; thereby implying a simple rank one feature matrix. To obtain this result, our main technical contribution is to show that label noise SGD always minimizes the sharpness on the manifold of models with zero loss for two-layer networks. Along the way, we discover a novel property — a local geodesic convexity — of the trace of Hessian of the loss at approximate stationary points on the manifold of zero loss, which links sharpness to the geometry of the manifold. This tool may be of independent interest.

## 1 Introduction

Overparameterized neural networks trained by stochastic gradient descent (SGD) have demonstrated remarkable generalization ability. The emergence of this ability, even when the network perfectly fits the data and without any explicit regularization, still remains a mystery (Zhang et al., 2017). A line of recent works have attempted to provide a theoretical basis for such observations by showing that *SGD has an implicit bias toward functions with "low complexity" among all possible networks with zero training loss* (Neyshabur et al., 2017; Soudry et al., 2018; Gunasekar et al., 2017; Li et al., 2018; Gunasekar et al., 2018; Woodworth et al., 2020; Li et al., 2019; HaoChen et al., 2021; Pesme et al., 2021). In particular, it is noted that SGD tends to pick simpler features to fit the data over more complex ones (Hermann & Lampinen, 2020; Kalimeris et al., 2019; Neyshabur et al., 2014; Pezeshki et al., 2021). This tendency is typically referred to as *simplicity bias*. Notably, Huh et al. (2021) empirically observes that the feature matrix on the training set tends to become low-rank and that this property is amplified in deeper networks.

On the other hand, recently, a different form of implicit bias of SGD based on the sharpness of loss landscape has garnered interest. The study of flatness of the loss landscape and its positive correlation with generalization is not new (e.g. (Hochreiter & Schmidhuber, 1997)), and has been supported by a plethora of empirical and theoretical evidence (Keskar et al., 2016; Dziugaite & Roy, 2017; Jastrzebski et al., 2017; Neyshabur et al., 2017; Wu et al., 2018; Jiang et al., 2019; Blanc et al., 2019; Wei & Ma, 2019a;b; HaoChen et al., 2020; Foret et al., 2021; Damian et al., 2021; Li et al., 2021; Ma & Ying, 2021; Ding et al., 2022; Nacson et al., 2022; Wei et al., 2022; Lyu et al., 2022; Norton & Royset, 2021). It has been suggested that, after reaching zero loss, neural networks trained using some variants of SGD are biased toward regions of the loss landscape with low sharpness (Blanc et al., 2019; Damian et al., 2021; Li et al., 2021; Arora et al., 2022; Lyu et al., 2022; Li et al., 2022a; Wen et al., 2022; Bartlett et al., 2022). In particular, Blanc et al. (2020) discovered the sharpness-reduction implicit bias of a variant of SGD called *label noise SGD*, which explicitly adds noise to the labels. They proved that SGD locally diverges from points that are not

stationary for the trace of the Hessian of the loss after fitting the training data. Along this line, Li et al. (2021) show that in the limit of step size going to zero, label noise SGD converges to a gradient flow according to the trace of Hessian of the loss after reaching almost zero loss. This flow can be viewed as a Riemannian gradient flow with respect to the sharpness on the manifold of zero loss, *i.e.*, the flow following the gradient of sharpness projected onto the tangent space of the manifold. Furthermore, a standard Bayesian argument based on minimum description length suggests such flat minimizers correspond to "simple" models.

From a conceptual point of view, both simplicity bias and sharpness minimization bias suggest that SGD learns *simple* solutions; however, the nature of "simplicity" is different. Simplicity bias is defined in the feature space, meaning the model learns simple features, *e.g.*, low-rank features, but low sharpness is defined in the parameter space, meaning the loss landscape around the learned parameter is flat, such that the model is resilient to random parameter perturbations and admits a short and simple description from a Bayesian perspective. It is thus an interesting question whether these two notions of "simplicity" for neural networks have any causal relationship. Existing works that attempt to study this connection are restricted to somewhat contrived settings (e.g. Blanc et al. (2020); Shallue et al. (2018); Szegedy et al. (2016); Shallue et al. (2018); HaoChen et al. (2021)). For instance, Blanc et al. (2020) study 1-D two-layer ReLU networks and two-layer sigmoid networks trained on a single data point, where they show the stationary points of the trace of Hessian on the manifold of zero loss are solutions that are simple in nature. They further prove that label noise SGD locally diverges from non-stationary points. As another example, Li et al. (2021) and Vivien et al. (2022) independently show that label noise SGD can recover the sparse ground truth, but only for a simple overparameterized quadratic linear model. In this paper, we study the following fundamental question:

> *(1) Is there a non-linear general setting where the sharpness-reduction implicit bias provably implies simplicity bias?*

However, even if we can show that a model with minimum flatness is a simple model, a key open question is whether the sharpness-reduction implicit bias, characterized by the riemannian gradient flow of sharpness in (Li et al., 2021), even successfully minimizes the sharpness in the first place. The convergence of the gradient flow so far is only known in very simple settings, *i.e.*, quadratically overparametrized linear nets Li et al. (2021). In this regard, a central open question in the study of sharpness minimization for label noise SGD is:

> *(2) Does label noise SGD converge to the global minimizers of the sharpness on the manifold of zero loss, and if it does, how fast does it converge?*

In this paper, we show sharpness regularization provably lead to simplicity bias for two-layer neural networks with certain activation functions. We further prove that the Riemannian gradient flow on the manifold linearly converges to the global minimizer of the sharpness *for all initialization*. To our best knowledge, this is the first global linear convergence result for the gradient flow of trace of Hessian on manifolds of minimizers.

More formally, we consider the mean squared loss $\mathcal{L}(\theta) = \sum_{i=1}^{n}(\sum_{j=1}^{m}\phi(\theta_j^\top x_i) - y_i)^2$ for a two-layer network model where the weights of the second layer are fixed to one and $\{x_i, y_i\}_{i=1}^{n}$ are the training dataset. [1] We consider label noise SGD on $\mathcal{L}$, that is, running gradient descent on the loss $\sum_{i=1}^{n}(\sum_{j=1}^{m}\phi(\theta_j^\top x_i) - y_i + \zeta_{t,i})^2$ at each step $t$, in which independent Gaussian noise $\zeta_{t,i} \sim \mathcal{N}(0, \sigma^2)$ is added to the labels. The following is the primary main result of our paper, which holds under Assumptions 1, 2, 3) and for sufficiently small learning rate.

**Theorem 1** (Convergence to simple solutions from any initialization). *Given any $\epsilon > 0$ and arbitrary initial parameter $\theta[0]$, for label noise SGD with any noise variance $\sigma^2 > 0$ initialized at $\theta[0]$, there is a sufficiently small learning rate $\eta$ such that running label noise SGD with step size $\eta$ and $T = \tilde{O}(\log(1/\epsilon) \cdot \eta^{-2}\sigma^{-2})$, with probability at least $1 - \epsilon$,*

1. *$\mathcal{L}(\theta[T]) \leq \epsilon$;*

2. *$\mathrm{Tr}(D^2\mathcal{L}(\theta[T])) \leq \inf_{\theta:\mathcal{L}(\theta)=0} \mathrm{Tr}(D^2\mathcal{L}(\theta)) + \epsilon$;*

---

[1]We discuss how our result generalizes to the case of unequal weights at the end of Section 4.2.

3. *For all neurons $\theta_j[T]$ and all data points $x_i$, $\left|\theta_j[T]^\top x_i - \phi^{-1}(y_i/m)\right| \leq \epsilon$,*

*where $D^2\mathcal{L}$ is the Hessian of the loss and we hide constants about loss $\mathcal{L}$ and initialization $\theta[0]$ in $O(\cdot)$.*

Informally, Theorem 1 states that, first, label noise SGD reaches zero loss and then (1) successfully minimizes the sharpness to its global optima (2) and more importantly, it recovers the simplest possible solution that fits the data, in which the feature matrix is rank one (see Appendix C.3 for proof details). In particular, Theorem 1 holds for pure label noise SGD without any additional projection. Note that if we initialize the weights at zero, then because the update of label noise SGD is always in the span of the data points, the weights remain in the data span as well. On the other hand, property 3 in Theorem 1 states that the dot product of all the neurons to each single data point converge to each other, which means all the neurons collapse into the same vector. Our technical contributions cover the following two aspects:

- **Simplicity bias:** We prove that at every minimizer of the trace of Hessian, the pre-activation of different neurons becomes the same for every training data. In other words, all of the feature embeddings of the dataset with respect to different neurons collapse into a single vector, *e.g.*, see Figure 1a. This combined with our convergence result mathematically proves that the low-rank feature observation by (Huh et al., 2021) holds in our two-layer network setting trained by sharpness minimization induced regularizer algorithms, such as label noise SGD and 1-SAM.

- **Convergence analysis:** Under an additional normality assumption on the activation (Assumption 2), we show that the gradient flow of the trace of Hessian converges exponentially fast to the global optima on the manifold of zero loss, and as a result, label noise SGD with a sufficiently small step size successfully converges to the global minimizer of trace of Hessian among all the models achieving zero loss. Importantly, we do not assume additional technical conditions (such as PL inequality and non-strict saddle property used in prior works) about the landscape of the loss or its sharpness, and our constants are explicit and only depend on the choice of the activation and coherence of the data matrix. Moreover, our convergence results hold in the strong sense i.e., the convergence holds for the last iterations of label noise SGD.

  The novelty of our approach (after that we show it converges to zero loss) is that we characterize the convergence on the manifold of zero loss in two phases (1) the first phase where the algorithm is far from stationary points, trace of Hessian decreases with a constant rate and there is no convexity in trace of Hessian, (2) the second phase where the algorithm gets close to stationarity, we prove a novel g-convexity property of trace of Hessian which holds only locally on the manifold, but we show implies an exponentially fast convergence rate by changing the Lyapunov potential from value of trace of Hessian to the norm squared of its gradient on the manifold. We further prove that approximate stationary points are physically close to a global optimum via a semi-monotonicity property.

Interestingly, we observe this simplicity bias even beyond the regime of our theory; namely when the number of data points exceeds the ambient dimension, instead of a single vector, the feature embeddings of the neurons cluster and collapse into a small set of vectors. (see Figure 1b)

## 2 RELATED WORK

**Implicit Bias of Sharpness Minimization.** Recent theoretical investigations, including those by Blanc et al. (2019), Damian et al. (2021), and Li et al. (2021), have indicated that Stochastic Gradient Descent (SGD) with label noise intrinsically favors local minimizers with smaller Hessian traces. This is under the assumption that these minimizers are connected locally as a manifold and these analyses focus on the final phase of training when iterates are close to the manifold of minimizes. The effect of label noise or mini-batch noise in the central phase of training, *i.e.*, when the loss is still substantially above zero is more difficult and is only known for simple quadratically over-parametrized linear models (Vivien et al., 2022; Andriushchenko et al., 2023b). Further, Arora et al. (2022) demonstrated that normalized Gradient Descent (GD) intrinsically penalizes the Hessian's largest eigenvalue. Ma et al. (2022) proposes that such sharpness reduction phenomena could also be triggered by a multi-scale loss landscape. In the context of scale-invariant loss functions, Lyu et al. (2022) found that GD with weight decay implicitly reduces the spherical sharpness, defined as the Hessian's largest eigenvalue evaluated at the normalized parameter.

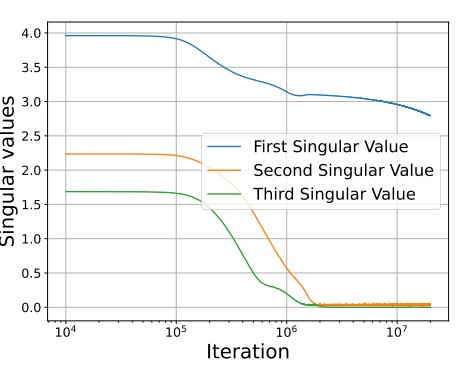

(a) The rank of the feature matrix and weight matrix go to one as Theorem 1 predicts in the high dimensional setting $d > n$.

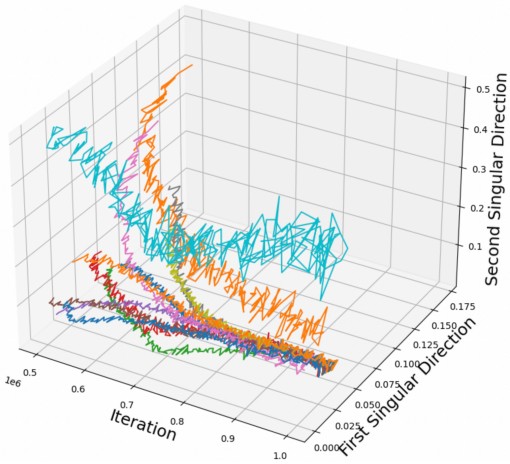

(b) The feature embeddings of neurons collapse into clusters in the low dimensional regime $d < n$. The plot visualizes this phenomenon using the first and second principal components of the feature embedding $\Theta^* X$.

Another line of research examines the sharpness minimization effect of large learning rates, assuming that (stochastic) gradient descent converges at the end of training. This analysis primarily hinges on the concept of linear stability, as referenced in works by Wu et al. (2018), Cohen et al. (2021), Ma & Ying (2021), and Cohen et al. (2022). More recent theoretical analyses, such as those by Damian et al. (2022) and Li et al. (2022b), suggest that the sharpness minimization effect of large learning rates in gradient descent does not necessarily depend on the convergence assumption and linear stability. Instead, they propose a four-phase characterization of the dynamics in the so-called Edge of Stability regime, as detailed in Cohen et al. (2021).

Sharpness-aware Minimization (SAM, Foret et al. (2021)) is a highly effective regularization method that improves generalization by penalizing the sharpness of the landscape. SAM was independently developed in (Zheng et al., 2021; Norton & Royset, 2021). Recently, Wen et al. (2022); Bartlett et al. (2022) proved that SAM successfully minimizes the worst-direction sharpness under the assumption that the minimizers of training loss connect as a smooth manifold. Wen et al. (2022) also analyze SAM with batch size 1 for regression problems and show that the implicit bias of SAM becomes penalizing average-direction sharpness, which is approximately equal to the trace of the Hessian of the training loss. Andriushchenko et al. (2023a) observe that the SAM update rule for ReLU networks is locally biased toward sparsifying the features leading to low-rank features. They further confirm empirically that deep networks trained by SAM are biased to produce low-rank features.

## 3 MAIN RESULTS

**Problem Setup.** In this paper, we focus on the following two-layer neural network model:

$$r_{\theta,\text{NN}}(x) \triangleq \sum_{j=1}^{m} \phi(\theta_j^\top x). \tag{1}$$

Here $\theta = (\theta_1, \ldots, \theta_m)$ and $\phi$ are the set of parameters and activation of the neural network, respectively. As illustrated in equation 1, we assume that all the second-layer weights are equal to one. At the end of section 4.2 we discuss how our approach in Theorem 2 generalizes to any fixed choice of weights in the second layer. We denote the matrix of the weights by $\Theta \in \mathbb{R}^{m \times d}$, whose $j^{\text{th}}$ row is $\theta_j$. Given a training dataset $\{(x_i, y_i)\}_{i=1}^{n}$, we study the mean squared loss

$$\mathcal{L}(\theta) \triangleq \sum_{i=1}^{n} (r_{\theta,\text{NN}}(x_i) - y_i)^2. \tag{2}$$

For simplicity, we assume that the data points have unit norm, i.e. $\forall i \in [n]$, $\|x_i\| = 1$. Let $\mathcal{M}$ denote the zero level set of $\mathcal{L}$: $\mathcal{M} \triangleq \{\theta \in \mathbb{R}^{md}\,\big|\,\mathcal{L}(\theta) = 0\}$. As we show in Lemma 16 in the Appendix, $\mathcal{M}$ is well-defined as a manifold due to a mild non-degeneracy condition implied by Assumption 1. We equip $\mathcal{M}$ with the standard Euclidean metric in $\mathbb{R}^{md}$.

The starting point of our investigation is the work Li et al. (2021), which characterizes the minimizer of the trace of the Hessian of the loss, $\mathrm{Tr}D^2\mathcal{L}$, as the implicit bias of SGD by showing that, in the limit of step size going to zero, label noise SGD evolves on the manifold of zero loss according to the following deterministic gradient flow:

$$\frac{d}{dt}\theta(t) \triangleq -\nabla \mathrm{Tr}D^2\mathcal{L}(\theta(t)). \tag{3}$$

Here, $\nabla$ is the Riemannian gradient operator on the manifold $\mathcal{M}$, which is the projection of the normal Euclidean gradient onto the tangent space of $\mathcal{M}$ at $\theta(t)$. Note that starting from a point $\theta(0)$ on $\mathcal{M}$, $\theta(t)$ remains on $\mathcal{M}$ for all times $t$ (for a quick recap on basic notions in Differential Geometry such as covariant derivative and gradient on a manifold, we refer the reader to Appendix J.) On a manifold, similar to Euclidean space, having a zero gradient $\|\nabla \mathrm{Tr}D^2\mathcal{L}(\theta^*)\| = 0$ (the gradient defined on the manifold) at some points $\theta^*$ is a necessary condition for being a local minimizer, or equivalently the projection of Euclidean gradient onto the tangent space at $\theta^*$ has to vanish. More generally, we call $\theta$ an $\epsilon$-stationary point on $\mathcal{M}$ if $\|\nabla \mathrm{Tr}D^2\mathcal{L}(\theta)\| \leq \epsilon$.

## 3.1 Characterization of Stationary Points

Before characterizing the stationary points of $\mathrm{Tr}D^2\mathcal{L}$, we introduce an important assumption on the activation function, namely the convexity and positivity of its derivative.

**Assumption 1** (Strict positivity and convexity of $\phi'$)**.** *For all $z \in \mathbb{R}$, $\phi'(z) \geq \varrho_1 > 0, \phi'''(z) \geq \varrho_2 > 0$.*

For example, Assumption 1 is satisfied for $\phi(x) = x^3 + \nu x$ with constants $\varrho_1 = \nu, \varrho_2 = 1$. In Lemma 15, given the condition $\phi''' > 0$ we show that $\theta(t)$ remains in a bounded domain along the gradient flow, which means having the weaker assumption that $\phi', \phi''' > 0$ automatically implies Assumption 1 for some positive constants $\varrho_1$ and $\varrho_2$. Another activation of interest to us that happens to be extensively effective in NLP applications is the cube ($x^3$) activation Chen & Manning (2014). Although $x^3$ does not satisfy the strict positivity required in Assumption 1, we separately prove Theorem 2 for $x^3$ (see Lemma 18 in the appendix.)

Next, let $X \triangleq \left(x_1\big|\ldots\big|x_n\right)$ be the data matrix. We make a coherence assumption on $X$ which requires the input dimension $d$ to be at least as large as the number of data points $n$.

**Assumption 2** (Data matrix coherence)**.** *The data matrix $X$ satisfies $X^\top X \geq \mu I$.*

Assumption 1 implies that $\mathcal{M}$ is well-defined as a manifold (see Lemma 16 for a proof). Under aforementioned Assumptions 1 and 2, we show that the trace of the Hessian regularizer has a unique stationary point on the manifold.

**Theorem 2** (First-order optimal points)**.** *Under Assumptions 1 and 2, the first order optimal points and global optimums of $\mathrm{Tr}D^2\mathcal{L}$ on $\mathcal{M}$ coincide and are equal to the set of all $\theta^* = \left[\theta_j^*\right]_{j=1}^m$ such that for all $i \in [n]$ and $j \in [m]$ satisfy the following:*

$$\theta_j^{*\top} x_i = \phi^{-1}(y_i/m). \tag{4}$$

Note that instead of Assumption 2, just having the linear independence of $\{x_i\}_{i=1}^n$ suffices to prove Theorem 2, as pointed out in section 4.2. In particular, Theorem 2 means that the feature matrix is rank one at a global optimum $\theta^*$, which together with our main convergence result Theorem 1 proves the low-rank bias conjecture proposed by (Huh et al., 2021) in the two-layer network setting for label noise SGD and 1-SAM. Notice the $1/m$ effect of the number of neurons on the global minimizers of the implicit bias in equation 4. While Theorem 2 concerns networks with equal second-layer weights, the case of unequal weights can be analyzed similarly, as we discuss after the proof of Theorem 2 in Section 4.2.

## 3.2 CONVERGENCE RATE

Next, to state our convergence result, we introduce the key $\beta$-normality assumption, under which we bound the rate of convergence of the trace of Hessian to its global minimizer.

**Assumption 3** ($\beta$-normality). *For all $z \in \mathbb{R}$ the second derivative of $\phi(z)$ can be bounded by the first and third derivatives as*

$$\beta \phi''(z) \leq \phi'^2(z)\phi'''(z).$$

An example class of activations that satisfies both Assumptions 1 and 3 is of the form $\phi(z) = z^{2k+1} + \nu z$ for $\nu > 0$, which is well-known in the deep learning theory literature Li et al. (2018); Jelassi et al. (2022); Allen-Zhu & Li (2020b); Woodworth et al. (2020). Notably, these activations satisfy Assumption 3 with normality coefficient $\beta = \min\{\frac{1}{(2k+1)^2(2k-1)}, \nu^2\}$.

Under Assumptions 1, 2, and 3, we show that $\theta(t)$ converges to $\theta^*$ exponentially fast in Theorem 3. This results from strong local g-convexity of trace of Hessian for approximate stationary points (see Lemmas 4 and 10), plus a semi-monotonicity property for the trace of Hessian (Lemma 6).

**Theorem 3** (Convergence of the gradient flow). *Consider the limiting flow of Label noise SGD on the manifold of zero loss, which is the gradient flow in equation 3. Then, under Assumptions 1,2, and 3,*

> *(C.1) the gradient flow reaches $\epsilon$-stationarity for all times*
>
> $$t \geq \frac{\text{Tr}D^2\mathcal{L}(\theta(0))}{\mu\beta^2} + \frac{\log(\beta^2/(\varrho_1^2\varrho_2^2\epsilon^2) \vee 1)}{\varrho_1\varrho_2\mu}.$$

> *(C.2) For all $j \in [m]$ and $i \in [n]$, the dot product of the $j$th neuron to the $i$th data point becomes $\epsilon$-close to that of any global optimum $\theta^*$:*
>
> $$\left|\theta_j(t)^\top x_i - \theta_j^{*\top}x_i\right| \leq \epsilon, \tag{5}$$
>
> *where $\theta^*$ is defined in Theorem 2.*

The formal proof of Theorem 3 is given in Section C.3. We sketch its proof in Section 4.3. Note that the rate is logarithmic in the accuracy parameter $\epsilon$, i.e., the flow has exponentially fast convergence to the global minimizer.

## 4 PROOF SKETCHES

Before going into the technical details of the proofs, we discuss some necessary background and additional notation used in the proofs.

## 4.1 ADDITIONAL NOTATION

We use $f_i(\theta)$ to denote the output of the network on the $i$th input $x_i$, i.e.

$$f_i(\theta) \triangleq r_{\theta,\text{NN}}(x_i). \tag{6}$$

We use $f(\theta) \triangleq (f_1(\theta), \ldots, f_n(\theta))$ to denote the array of outputs of the network on $\{x_i\}_{i=1}^n$. We use $D$ for the Euclidean directional derivative or Euclidean gradient, and $\nabla$ for the covariant derivative on the manifold or the gradient on the manifold. We denote the Jacobian of $f$ at point $\theta$ by $Df(\theta)$ whose $i$th row is $Df_i(\theta)$, the Euclidean gradient of $f_i$ at $\theta$. Recall the definition of the manifold of zero loss as the zero level set of the loss $\mathcal{L}$, which is the intersection of zero level sets of functions $f_i, \forall i \in [m]$: $\mathcal{M} = \left\{\theta \in \mathbb{R}^{md} | \forall i \in [n], f_i(\theta) = y_i\right\}$. The tangent space $\mathcal{T}_\theta(\mathcal{M})$ of $\mathcal{M}$ at point $\theta$ can be identified by the tangents to all curves on $\mathcal{M}$ passing through $\theta$, and the normal space $\mathcal{T}_\theta^N(\mathcal{M})$ in this setting is just the orthogonal subspace of $\mathcal{T}_\theta(\mathcal{M})$. We denote the projection operators onto $\mathcal{T}_\theta(\mathcal{M})$ and $\mathcal{T}_\theta^N(\mathcal{M})$ by $P_\theta$ and $P_\theta^N$, respectively. For a set of vectors $\{v_i\}_{i=1}^n$ where for all $i \in [n]$, $v_i \in \mathbb{R}^d$, we use the notation $\left[v_i\right]_{i=1}^n$ to denote the vector in $\mathbb{R}^{nd}$ obtained by stacking vectors $\{v_i\}_{i=1}^n$.

## 4.2 PROOF SKETCH OF THEOREM 2

In this section, we describe the high-level proof idea of Theorem 2. Note that Theorem 2 characterizes the first order stationary points of $\text{Tr} D^2 \mathcal{L}$ on $\mathcal{M}$, i.e., points $\theta^*$ on $\mathcal{M}$ for which $\nabla \text{Tr} D^2 \mathcal{L}(\theta^*) = 0$. The starting point of the proof is the observation that the gradients $D f_i(\theta)$ form a basis for the normal space at $\theta$.

**Lemma 1** (Basis for the normal space). *For every $\theta$, the set of vectors $\{D f_i(\theta)\}_{i=1}^n$ form a basis for the normal space $\mathcal{T}_\theta^N(\mathcal{M})$ of $\mathcal{M}$ at $\theta$.*

For a stationary point $\theta^*$ with $\nabla F(\theta^*) = 0$, the Euclidean gradient at $\theta^*$ should be in the normal space $\mathcal{T}_{\theta^*}^N(\mathcal{M})$. But by Lemma 1, this means there exist coefficients $\{\alpha_i\}_{i=1}^n$ such that

$$D\text{Tr} D^2 \mathcal{L}(\theta^*) = \sum_{i=1}^n \alpha_i D f_i(\theta^*). \tag{7}$$

To further understand what condition equation 7 means for our two-layer network equation 1, we first state a result from prior work in Lemma 2 (see e.g. Blanc et al. (2020)) regarding the trace of the Hessian of the mean squared loss $\mathcal{L}$ for some parameter $\theta \in \mathcal{M}$ (see Section D.2 for a proof).

**Lemma 2.** *For the loss on the dataset defined in equation 2, for $\theta$ with $\mathcal{L}(\theta) = 0$, we have*

$$\text{Tr} D^2 \mathcal{L}(\theta) = \sum_{i=1}^n \|D f_i(\theta)\|^2.$$

Using Lemma 2, we can explicitly calculate the trace of Hessian for our two-layer network in equation 1.

**Lemma 3** (Trace of Hessian in two-layer networks). *For the neural network model equation 1 and the mean squared loss $\mathcal{L}$ in equation 2, we have*

$$\text{Tr} D^2 \mathcal{L}(\theta) = \sum_{i=1}^n \sum_{j=1}^m \phi'(\theta_j^\top x_i)^2. \tag{8}$$

Now we are ready to prove Theorem 2.

*Proof of Theorem 2.* From equation 7 and by explicitly calculating the gradients $D f_i(\theta^*)$'s using Lemma 3, we have

$$\left[ \sum_{i=1}^n 2\phi'(\theta_j^{*\top} x_i)\phi''(\theta_j^{*\top} x_i)x_i \right]_{j=1}^m = \sum_{i=1}^n \alpha_i \left[ \phi'(\theta_j^{*\top} x_i)x_i \right]_{j=1}^m.$$

But using our assumption that the data points $\{x_i\}_{i=1}^n$ are linearly independent, we have for all $i \in [n]$ and $j \in [m]$, $\phi''(\theta_j^{*\top} x_i) = \alpha_i$. Now, because $\phi'''$ is positive and $\phi''$ is strictly monotone, its inverse is well-defined:

$$\theta_j^{*\top} x_i = \nu_i \triangleq \phi''^{-1}(\alpha_i), \tag{9}$$

This implies $\phi(\theta_j^{*\top} x_i) = \phi(\nu_i)$. Therefore,

$$y_i = \sum_{j=1}^m \phi(\theta_j^{*\top} x_i) = m\phi(\nu_i). \tag{10}$$

But note that from strict monotonicity of $\phi$ from Assumption 1, we get that it is invertible. Therefore, equation 10 implies

$$\nu_i = \phi^{-1}(y_i/m), \quad \text{and} \quad \alpha_i = \phi''(\phi^{-1}(y_i/m)).$$

This characterizes the first order optimal points of $\text{Tr} D^2 \mathcal{L}$.

On the other hand, note that $\text{Tr} D^2 \mathcal{L} \geq 0$, so its infimum over $\mathcal{M}$ is well-defined. Moreover, the fact that the Jacobian $D f(\theta)$ is non-degenerate at all points $\theta \in \mathcal{M}$ (Lemma 17) implies that $\mathcal{M}$ is also topologically closed in $\mathbb{R}^{md}$, hence from the continuity of $\text{Tr} D^2 \mathcal{L}$ it achieves its infimum. But global minimizers are also stationary points and from equation 9 we see that all of the first order optimal points have the same value of $\text{Tr} D^2 \mathcal{L}$. Therefore, all first order optimal points are global optima as well, and they satisfy equation 9. □

**Results for General Second Layer Weights:** Note that if the weights of the second layer are fixed arbitrarily to $(w_j)_{j=1}^m$, then equation 9 simply turns into ${\theta_j^*}^\top x_i = \phi''^{-1}(\alpha_i/w_j)$; namely, the $\phi''$ image of the feature matrix $\Theta^* X$ is rank one. Interestingly, this shows that in the general case, the right matrix to look for obtaining a low-rank structure is the embedding of the feature matrix by the second derivative of the activation (for the cube activation this embedding is the feature matrix itself.) Next, we investigate the convergence of the gradient flow in equation 3 to a global optimum $\theta^*$.

### 4.3 PROOF SKETCH OF THEOREM 3

The first claim **(C.1)** of Theorem 3 is that after time $t \geq \tilde{\Omega}(\log(1/\epsilon))$, the gradient will always be smaller than $\epsilon$. It is actually trivial to prove that there exists some time at most $t = \tilde{O}(1/\epsilon^2)$ such that $\|\nabla \mathrm{Tr} D^2 \mathcal{L}(\theta(t))\| \leq \epsilon$ without any assumption, based on a standard argument which is a continuous version of descent lemma. Namely, if the gradient of $\mathrm{Tr} D^2 \mathcal{L}$ remains larger than some $\epsilon > 0$ along the flow until time $t > \mathrm{Tr} D^2 \mathcal{L}(\theta(0))/\epsilon^2$, then we get a contradiction:

$$\mathrm{Tr} D^2 \mathcal{L}(\theta(0)) - \mathrm{Tr} D^2 \mathcal{L}(\theta(t)) = \int_0^t \left\| \nabla \mathrm{Tr} D^2 \mathcal{L}(\theta(t)) \right\|^2 > \mathrm{Tr} D^2 \mathcal{L}(\theta(0)).$$

Our novelty here is a new characterization of the loss landscape under Assumptions 1, 2 (Lemma 4 proved in Appendix C.1).

**Lemma 4** (PSD Hessian when gradient vanishes). *Suppose that activation $\phi$ satisfies Assumptions 1, 2, and 3. Consider a point $\theta$ on the manifold where the gradient is small, namely $\|\nabla_\theta \mathrm{Tr} D^2 \mathcal{L}(\theta)\| \leq \sqrt{\mu}\beta$. Then, the Hessian of $\mathrm{Tr} D^2 \mathcal{L}$ on the manifold is PSD at point $\theta$, or equivalently $\mathrm{Tr} D^2 \mathcal{L}$ is locally g-convex on $\mathcal{M}$ around $\theta$.*

Lemma 4 implies that whenever the gradient is sufficiently small, the time derivative of the squared gradient norm will also be non-positive:

$$\frac{d}{dt}\|\nabla \mathrm{Tr} D^2 \mathcal{L}(\theta(t))\|^2 = -\nabla \mathrm{Tr} D^2 \mathcal{L}(\theta(t))^\top \nabla^2 \mathrm{Tr} D^2 \mathcal{L}(\theta(t)) \nabla \mathrm{Tr} D^2 \mathcal{L}(\theta(t)) \leq 0,$$

Therefore, once the gradient is sufficiently small, it will always remain small. In fact, we show that the Hessian of $\mathrm{Tr} D^2 \mathcal{L}$ on the manifold is strictly positive with a uniform $\varrho_1 \varrho_2 \mu$ lower bound in a suitable subspace which includes the gradient (Lemma 10). Then

$$\frac{d\|\nabla \mathrm{Tr} D^2 \mathcal{L}(\theta(t))\|^2}{dt} = -\nabla \mathrm{Tr} D^2 \mathcal{L}(\theta(t))^\top \nabla^2 \mathrm{Tr} D^2 \mathcal{L}(\theta(t)) \nabla \mathrm{Tr} D^2 \mathcal{L}(\theta(t)) \leq -\varrho_1 \varrho_2 \mu \|\nabla \mathrm{Tr} D^2 \mathcal{L}(\theta(t))\|^2,$$

which further implies a linear convergence of gradient norm:

$$\|\nabla \mathrm{Tr} D^2 \mathcal{L}(\theta(t))\|^2 \leq \|\nabla \mathrm{Tr} D^2 \mathcal{L}(\gamma(\theta(t_0)))\|^2 e^{-(t-t_0)\varrho_1 \varrho_2 \mu}.$$

Next, we state the high level ideas that we use to prove Lemma 4, which is the key to proving Theorem 3. Before that, we recall some necessary background from Differential Geometry.

**Computing the Hessian on $\mathcal{M}$.** To prove Lemma 4 we need to calculate the Hessian of $\mathrm{Tr} D^2 \mathcal{L}$ on the manifold. Note that the Hessian of a function $F$ on $\mathbb{R}^{md}$ can be defined as $D^2 F[u, w] = \langle D(DF(\theta))[u], w \rangle$ where $DF(\theta)$ is the usual Euclidean gradient of $F$ at $\theta$, and $D(.)[u]$ denotes directional derivative. To calculate the Hessian on the manifold, one needs to substitute the Euclidean gradient $DF(\theta)$ by the gradient $\nabla F(\theta)$ on the manifold. Moreover, $D(DF(\theta))[u]$ has to be substituted by the covariant derivative $\nabla_u \nabla F(\theta)$, a different differential operator than the usual derivative We recall the characterization of the covariant derivative as the projection of the conventional directional derivative onto the tangent space. For more background on covariant differentiation, we refer the reader to Appendix J.

**Fact 1.** *For vector fields $V, W$ on $\mathcal{M}$, we have $\nabla_V W(\theta) = P_\theta DW(\theta)[V]$.*

We also recall the definition of Hessian $\nabla^2 F$ on $\mathcal{M}$ based on Covariant derivative.

**Fact 2.** *The Hessian of $F$ at point $\theta$ on $\mathcal{M}$ is given by $\nabla^2 F(w, u) = \langle \nabla_w \nabla F, u \rangle$.*

We point out that on a general manifold the dot product $\langle,\rangle$ in Fact 2 is with respect to the metric of the manifold. However, in the case of a hypersurface $\mathcal{M} \subseteq \mathbb{R}^{md}$, the metric is the same as that of the Euclidean chart $\mathbb{R}^{md}$ that $\mathcal{M}$ is embedded in.

Next, we explicitly calculate the Hessian of $F \triangleq \mathrm{Tr}D^2\mathcal{L}$ on $\mathcal{M}$ in Lemma 8, which is the key in relating the norm of the gradient of $\mathrm{Tr}D^2\mathcal{L}$ to its Hessian in proving Lemma 4. exploiting the formula that we derive in Lemma 5 in Appendix C for the Hessian of a general smooth function $F$ over $\mathcal{M}$. Lemma 5 is proved in Appendix C.1.1.

**Lemma 5** (Hessian of the implicit regularizer on the manifold). *Recall that $\{Df_i(\theta)\}_{i=1}^n$ is a basis for the normal space $\mathcal{T}_\theta^N(\mathcal{M})$ according to Lemma 1. Let $\alpha' = (\alpha_i')_{i=1}^n$ be the coefficients representing $P_\theta^N(D(\mathrm{Tr}D^2\mathcal{L})(\theta)) \in \mathcal{T}_\theta^N(\mathcal{M})$ in the basis $\{Df_i(\theta)\}_{i=1}^n$, i.e.*

$$P_\theta^N\Big(D(\mathrm{Tr}D^2\mathcal{L})(\theta)\Big) = \sum_{i=1}^n \alpha_i' Df_i(\theta).$$

*Then, the Hessian of $\mathrm{Tr}D^2\mathcal{L}$ on $\mathcal{M}$ can be explicitly written (in the Euclidean chart $\mathbb{R}^{md}$) using $\alpha'$ as*

$$\nabla^2 \mathrm{Tr}D^2\mathcal{L}(\theta)[u,w] = D^2\mathrm{Tr}D^2\mathcal{L}(\theta)[u,w] - \sum_{i=1}^n \alpha_i' D^2 f_i(\theta)[u,w], \tag{11}$$

*for arbitrary $u, w \in \mathbb{R}^{md}$, where recall that $D^2$ denotes the normal Euclidean Hessian while we use $\nabla^2$ for the Hessian over the manifold.*

Observe in the formula of $\nabla^2\mathrm{Tr}D^2\mathcal{L}$ in equation 11 the first term is just the normal Euclidean Hessian of $\mathrm{Tr}D^2\mathcal{L}$ while we get the second "projection term" due to the difference of covariant differentiation with the usual derivative. Finally, to prove the second claim **(C.2)** of Theorem 3, we prove Lemma 6 in Appendix D.5 which shows that the small gradient norm of $\|\nabla\mathrm{Tr}D^2\mathcal{L}(\theta)\| \leq \delta$ implies the (approximate) alignment of features.

**Lemma 6** (Small gradient implies close to optimum). *Suppose $\|\nabla\mathrm{Tr}D^2\mathcal{L}(\theta)\| \leq \delta$. Then, for all $i \in [n], j \in [m], \left|\theta_j^\top x_i - \theta_j^{*\top} x_i\right| \leq \delta/(\sqrt{\mu}\varrho_1\varrho_2)$, for $\theta^*$ as defined in equation 4.*

## 5 CONCLUSION

In this paper, we take an important step toward understanding the implicit bias of label noise SGD with trace of Hessian induced regularizer for a class of two layer networks. We discover an intriguing relationship between this induced regularizer and the low-rank simplicity bias conjecture of neural networks proposed in Huh et al. (2021): we show that by initializing the neurons in the subspace of high-dimensional input data, all of the neurons converge into a single vector. Consequently, for the final model, (i) the rank of the feature matrix is effectively one, and (ii) the functional representation of the final model is *very simple*; specifically, its sub-level sets are half-spaces. To prove this, in spite of the lack of convexity, we uncover a novel structure in the landscape of the loss regularizer: the trace of Hessian regularizer becomes locally geodetically convex at points that are approximately stationary. Furthermore, in the limit of step size going to zero, we prove that label noise SGD or 1-SAM converge *exponentially fast* to a global minimizer. Generalizing the class of activations that enjoy fast convergence or proving the existence of fundamental barriers and handling the case of low-dimensional input data are interesting future directions. Based on the compelling empirical evidence from Huh et al. (2021) and our results for two-layer networks, we hypothesize that a low-rank simplicity bias can also be shown for deeper networks, possibly using the sharpness minimization framework.

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

## A    EXPERIMENTAL SETUP

We examine our theory on the convergence to a rank one feature matrix in Figure 1a via a synthetic experiment by considering a network with $m = 10$ neurons on ambient dimension $d = 3$ and $n = 3$ data points. We further pick learning rate $\eta = 0.05$ and noise variance $\sigma = 0.03$ for implementing label noise SGD. Each entry of the data points is generated uniformly on $[0, 1]$, which is the same data generating process in all the experiments. As Figure 1a shows, the second and third eigenvalues converge to zero which is predicted by Theorem 2. On the other hand, in a similar setting with the same learning rate $\eta = 0.05$ but larger $\sigma = 0.2$ we run a synthetic experiment with $d = 5$ this time smaller than the number of data points $n = 6$. Note that in this case, the feature matrix cannot be rank one anymore. However, one might guess that it still converges to something low-rank even though our theoretical result is shown only for the high dimensional case; in this case, the smallest possible rank to fit the data is indeed two, and surprisingly we see in Figure 2 that except the first and second eigenvalues of the feature matrix, the rest go to zero. To further test extrapolating our theory to the low dimensional case, this time we use a higher number of data points $n = 15$ with ambient dimension $d = 10$ and 30 neurons, with learning rate $\eta = 0, 1$ and noise variance $\sigma = 0.2$. We observe a very interesting property: even though the feature matrix is high rank in this case, the neurons tend to converge to one another especially when they are close, as if there is a hidden attraction force between them. To visualize our experimental discovery, in Figure 1b we plot ths first two principal components (with the largest singular values) of the feature vectors for each neuron, and observe how it changes while running label noise SGD. We further pick 12 neurons for which the first and second components are bounded in $[0, 0.1]$ and $[0, 0.5]$ respectively. As one can see in Figure 1b, most of the these principal components of neurons except possibly one outlier are converging to the same point. This is indeed the case for the feature vectors of neurons as well, namely, they collapse into clusters, but unfortunately, we only have two dimensions to visualize this, and we have picked the most two effective indicators, namely the first and second principal components.

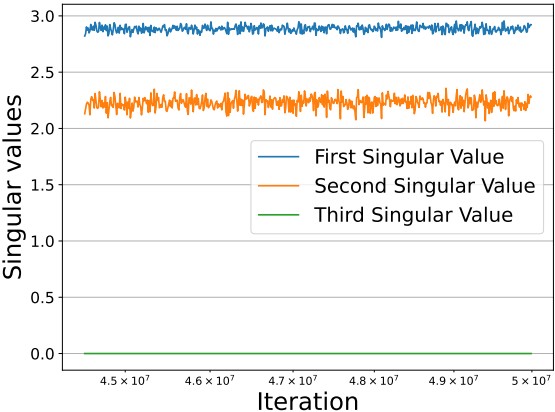

Figure 2: The rank of the feature matrix cannot converge to one when the ambient dimension $d = 5$ is smaller than the number of data points $n = 6$. However, as one can see in Figure 2 the rank in this case converges to two

## B    FURTHER DISCUSSION ON CONDITIONS THAT LEAD TO CONVERGENCE

In the literature, other conditions have been used to show convergence (rates) for nonconvex functions. We compare our analysis to those. In particular, in our analysis we prove and exploit a local g-convexity property.

**Local g-convexity vs PL inequality:**    If a function $F(\theta)$ satisfies a gradient dominance property such as PL (Polyak-Lojasiewicz) or KL inequalities, then one can show the convergence of the

gradient flow of $F$ to the global minimum. In particular, Damian et al. (2021) show convergence of at least one of the points along the trajectory of SGD somewhere close to a stationary point of $F$ for Label noise SGD, under some assumptions on the loss, including a KL inequality and Lipschitzness of the loss, its gradient, and Hessian. Given a PL or KL inequality, one can prove a bound on the shrinkage of the optimality gap.

Instead of gradient dominance property, we find the norm of the gradient to be a more effective potential for the landscape of the implicit bias in our setting. This is because once the flow reaches a region with small gradient, the local g-convexity property forces $\|\nabla F\|$ to become non-increasing, hence it gets trapped in the approximate stationary region. Note that this type of convergence is stronger than that in Damian et al. (2021), in the sense that it holds for any large enough time.

**Local g-convexity vs Strict Saddle property:** In this work, we can show the g-convexity property only locally at any approximate stationary point $\theta$. Hence, we cannot use the argument that $F$ is convex on the geodesic path from $\theta$ to the global optimum $\theta^*$, and the typical convergence analysis of gradient flow in the convex regime is ineffective here.

Our local g-convexity can be seen as a stronger property than the non-existence of non-strict saddle points, a popular assumption in the non-convex optimization and deep learning literature Jin et al. (2021); Allen-Zhu & Li (2020a). For instance, a specific variant of SGD whose noise is symmetric enough, or GD with random initialization, can escape strict saddle points in the non-convex landscape Lee et al. (2016); Jin et al. (2021). Yet, these results are not directly comparable to our setting. Namely, the result for convergence of GD in Lee et al. (2016) (1) uses dynamical systems theory and is inherently asymptotic, (2) needs the objective to be gradient Lipschitz, and (3) crucially depends on the random initialization. Rates for convergence for SGD with specific noise Jin et al. (2021) are also typically slow, the convergence is for the best point along the sequence and not for the last time/iterate, and they require Lipschitzness of the gradient. In fact, Kornowski & Shamir (2021) argue that in the non-smooth case, even finding $\epsilon$-stationary points in general is impossible. Finally, these frameworks have not yet been developed over a manifold which is the focus of this work. In contrast, our argument based on local g-convexity is simple and simply uses the fact that the norm of the gradient becomes non-increasing at approximate stationary points.

## C  FULL PROOF OF THEOREM 3

In order to prove Theorem 3, we take these steps:

- In Section C.1, we show that the norm of the gradient of $\mathrm{Tr}D^2\mathcal{L}$ becomes decreasing at approximate stationary points in Lemma 7. To this end, we show that under Assumptions 1, 2, 3, at each point $\theta \in \mathcal{M}$, we either have a large enough gradient or we are in a neighborhood of the global optima of the trace of Hessian in which trace of Hessian becomes g-convex as stated in Lemma 4.

- In Section C.2, Lemma 9, we show that the norm of the gradient of $\mathrm{Tr}D^2\mathcal{L}$ decays exponentially fast under Assumptions 1, 2, and 3. To show this, we prove Lemma 10 in which we show under Assumptions 1, 2, and 3 the Hessian of $\mathrm{Tr}D^2\mathcal{L}$ on $\mathcal{M}$ becomes positive definite at approximate stationary points.

- Finally, we combine Lemmas 7, 9, and 6 to show Theorem 3 in Appendix C.3.

### C.1  LOCAL G-CONVEXITY AT APPROXIMATE STATIONARY POINTS

In this section we prove Lemma 7.

**Lemma 7** (PSD Hessian when gradient vanishes)**.** *The norm of the gradient $\|\nabla \mathrm{Tr}D^2\mathcal{L}(\theta(t))\|$ on the gradient flow 3 becomes decreasing with respect to $t$ at $\sqrt{\mu}\beta$-approximate stationary points, i.e. when $\|\nabla \mathrm{Tr}D^2\mathcal{L}(\theta(t))\| < \sqrt{\mu}\beta$.*

To prove Lemma 7, we prove Lemma 4 in the rest of this section. Note that from the continuity of $\|\nabla \mathrm{Tr}D^2\mathcal{L}(\theta)\|$, we see that the Hessian of the implicit regularizer $\mathrm{Tr}D^2\mathcal{L}$ is PSD in a neighborhood of a $\sqrt{\mu}\beta/2$-stationary point $\theta \in \mathcal{M}$. This implies that $\mathrm{Tr}D^2\mathcal{L}$ is g-convex in a neighborhood of such $\theta$. We state a short proof of this in Lemma 19 in the appendix. Even though under Assumption 1

we get that $\text{Tr}D^2\mathcal{L}$ has a convex extension in $\mathbb{R}^{md}$, it is not necessarily g-convex over the manifold. This is because the Hessian over the manifold is calculated based on covariant differentiation, which is different from the normal Euclidean derivative. In particular, for a hypersurface $\mathcal{M}$ in $\mathbb{R}^{md}$, the covaraint derivative is obtained by taking the normal Euclidean derivative from the tangent of a curve on $\mathcal{M}$, then projecting the resulting vector back onto the tangent space. This projecting back operation in the definition of Covariant derivative results in an additional "projection term" in the algebraic formula of the Hessian of $\text{Tr}D^2\mathcal{L}$ on $\mathcal{M}$, in addition to its normal Euclidean Hessian. The inversion in the projection operator makes it fundamentally difficult to derive good estimates on the projection term that are relatable to the Euclidean Hessian of $\text{Tr}D^2\mathcal{L}$. Nonetheless, we discover a novel structure in $\mathcal{M}$ which is that even though we cannot control the projection term in the Hessian of $\text{Tr}D^2\mathcal{L}$ globally, but for $\epsilon$-stationary points $\theta$ on the manifold, we can relate it to the approximate stationarity condition (equivalently approximate KKT condition) at $\theta$. This relation enables us to control the projection term and show that the Hessian of $\text{Tr}D^2\mathcal{L}$ on $\mathcal{M}$ is indeed PSD at approximate stationary points. To prove Lemma 4, first we calculate the formula for the Hessian of an arbitrary function $F$ on a hypersurface $\mathcal{M}$ embedded in $\mathbb{R}^{md}$, in Lemma 8. We then use this formula to calculate the Hessian of $\text{Tr}D^2\mathcal{L}$ on $\mathcal{M}$ in Lemma 5. Finally we use this formula to prove Lemma 4 in Section 4.3.

Now we derive a general formula for the Hessian of any smooth function $F$ on the manifold in Lemma 8, proved in Appendix D.3.

**Lemma 8** (Hessian formula on a submanifold of $\mathbb{R}^{md}$). *For any smooth function $F$ defined on $\mathbb{R}^{md}$, the Hessian of $F$ on $\mathcal{M}$ is given by:*

$$\nabla^2 F(\theta)[u, w] = D^2 F[u, w] - D^2 f(\theta)[u, w]^\top (Df(\theta)Df(\theta)^\top)^{-1} Df(\theta)(DF(\theta)),$$

*for $u, w \in \mathcal{T}_\theta(\mathcal{M})$.*

Next, we prove Lemma 5, exploiting the formula that we derived in Lemma 8 for the Hessian of a general smooth function $F$ over $\mathcal{M}$.

### C.1.1 Proof of Lemma 5

*Proof.* Using the formula for the Hessian of a function over the manifold in Lemma 5, we have

$$\nabla^2 \text{Tr}D^2\mathcal{L}(\theta)[u, w]$$
$$= D^2 \text{Tr}D^2\mathcal{L}(\theta)[u, w] - D^2 f(\theta)[u, w]^\top (Df(\theta)Df(\theta)^\top)^{-1} Df(\theta)(D\text{Tr}D^2\mathcal{L}(\theta))$$
$$= D^2 \text{Tr}D^2\mathcal{L}(\theta)[u, w]$$
$$\quad - D^2 f(\theta)[u, w]^\top (Df(\theta)Df(\theta)^\top)^{-1} Df(\theta)\Big(\nabla_\theta \text{Tr}D^2 L(\theta) + 2\sum_{i=1}^n \alpha_i' Df_i(\theta)\Big)$$
$$\overset{equation\ 12}{=} D^2 \text{Tr}D^2\mathcal{L}(\theta)[u, w] - 2D^2 f(\theta)[u, w]^\top (Df(\theta)Df(\theta)^\top)^{-1} Df(\theta)\Big(\sum_{i=1}^n \alpha_i' Df_i(\theta)\Big)$$

$$\tag{12}$$

$$= D^2 \text{Tr}D^2\mathcal{L}[u, w] - 2D^2 f(\theta)[u, w]^\top (Df(\theta)Df(\theta)^\top)^{-1} (Df(\theta)Df(\theta)^\top)\alpha'$$
$$= D^2 \text{Tr}D^2\mathcal{L}(\theta)[u, w] - 2\sum_{i=1}^n \alpha_i' D^2 f_i(\theta)[u, w],$$

where equality equation 12 follows from the fact that $\nabla_\theta (\text{Tr}D^2\mathcal{L}) \in \mathcal{T}_\theta(\mathcal{M})$, which means it is orthogonal to $Df_i(\theta)$ for all $i \in [n]$. $\square$

Using Lemma 5, we prove Lemma 4. We restate Lemma 4 here for the convenience of the reader.

**Lemma** (see Lemma 4). *Suppose that activation $\phi$ satisfies Assumptions 1, 2, 3. Consider a point $\theta$ on the manifold where the gradient is small, namely $\|\nabla_\theta \text{Tr}D^2\mathcal{L}(\theta)\| \leq \sqrt{\mu}\beta$. Then, the Hessian of $\text{Tr}D^2\mathcal{L}$ on the manifold is PSD at point $\theta$, or equivalently $\text{Tr}D^2\mathcal{L}$ is locally g-convex on $\mathcal{M}$ around $\theta$.*

### C.1.2 Proof of Lemma 4

*Proof.* Suppose $\|\nabla_\theta \mathrm{Tr} D^2 \mathcal{L}(\theta)\| = \delta$. Write the projection of the Euclidean gradient $D(\mathrm{Tr} D^2 \mathcal{L})(\theta)$ onto the normal space $\mathcal{T}_\theta^N(\mathcal{M})$ in the row space of $Df(\theta)$, i.e. there exists a vector $\alpha' = (\alpha_i')_{i=1}^n$ such that

$$P_\theta^N [D(\mathrm{Tr} D^2 \mathcal{L})(\theta)] \triangleq 2 \sum_{i=1}^n \alpha_i' Df_i(\theta),$$

where $P_\theta^N$ is the projection onto the normal space at point $\theta$ on the manifold. Then, if we decompose $D(\mathrm{Tr} D^2 \mathcal{L})(\theta)$ into the part in the tangent space and the part in the normal space, because the part on the tangent space is the gradient of $\mathrm{Tr} D^2 \mathcal{L}$ on the manifold, we have

$$D(\mathrm{Tr} D^2 \mathcal{L})(\theta) = \nabla_\theta \mathrm{Tr} D^2 L(\theta) + 2 \sum_{i=1}^n \alpha_i' Df_i(\theta).$$

Now from the assumption that the norm of $\nabla \mathrm{Tr} D^2 \mathcal{L}(\theta)$ is at most $\delta$ and using the formula for $\mathrm{Tr} D^2 \mathcal{L}$ in Lemma 3, we have

$$\delta^2 = \left\| \nabla_\theta \mathrm{Tr} D^2 \mathcal{L}(\theta) \right\|^2 = \left\| D(\mathrm{Tr} D^2 \mathcal{L})(\theta) - 2 \sum_{i=1}^n \alpha_i' \left[ \phi'(\theta_j^\top x_i) x_i \right]_{j=1}^m \right\|^2$$

$$= 4 \sum_{j=1}^m \left\| \sum_{i=1}^n (\phi''(\theta_j^\top x_i) - \alpha_i') \phi'(\theta_j^\top x_i) x_i \right\|^2,$$

which implies

$$4\mu \sum_{i=1}^n (\phi''(\theta_j^\top x_i) - \alpha_i')^2 \phi'(\theta_j^\top x_i)^2 \le 4 \left\| \sum_{i=1}^n (\phi''(\theta_j^\top x_i) - \alpha_i') \phi'(\theta_j^\top x_i) x_i \right\|^2 \le \delta^2.$$

This further implies $\forall i \in [n], j \in [m]$:

$$|(\phi''(\theta_j^\top x_i) - \alpha_i') \phi'(\theta_j^\top x_i)| \le \delta/(2\sqrt{\mu}). \tag{13}$$

Now using the $\beta$-normality of $\phi$, the fact that $\delta/\sqrt{\mu} \le \beta$, and the positivity of $\phi'$ and $\phi'''$, we have for all $i \in [n]$ and $j \in [m]$,

$$2 \left| \phi''(\theta_j^\top x_i)(\alpha_i' - \phi''(\theta_j^\top x_i)) \right| \le \phi'(\theta_j^\top x_i) \phi'''(\theta_j^\top x_i). \tag{14}$$

On the other hand, using Lemma 5, we can explicitly calculate the Hessian of $\mathrm{Tr} D^2 \mathcal{L}$ as

$$\nabla^2 \mathrm{Tr} D^2 \mathcal{L}(\theta)[u, u] = D^2 \mathrm{Tr} D^2 \mathcal{L}(\theta)[u, u] - 2 \sum_{i=1}^n \alpha_i' D^2 f_i(\theta)[u, u].$$

Now using the formula for the Euclidean Hessian of $\mathrm{Tr} D^2 \mathcal{L}$ and $f_i$'s in Lemmas 13 and 14, we get

$$\nabla^2 \mathrm{Tr} D^2 \mathcal{L}(\theta)[u, u] = \sum_{j=1}^m \sum_{i=1}^n \left( 2\phi'(\theta_j^\top x_i) \phi'''(\theta_j^\top x_i) + 2\phi''(\theta_j^\top x_i)^2 \right)(x_i^\top u_j)^2$$

$$- \sum_{j=1}^m \sum_{i=1}^n 2\alpha_i' \phi''(\theta_j^\top x_i)(x_i^\top u_j)^2$$

$$= \sum_{j=1}^m \sum_{i=1}^n \left( 2\phi'(\theta_j^\top x_i) \phi'''(\theta_j^\top x_i) + 2\phi''(\theta_j^\top x_i)(\phi''(\theta_j^\top x_i) - \alpha_i') \right)(x_i^\top u_j)^2$$

$$\ge \sum_{j=1}^m \sum_{i=1}^n \phi'(\theta_j^\top x_i) \phi'''(\theta_j^\top x_i)(x_i^\top u_j)^2 \ge 0, \tag{15}$$

where the second to last inequality follows from Equation equation 14. Hence, the Hessian over the manifold is PSD at $\sqrt{\mu}\beta$-stationary points. $\square$

Based on Lemma 4, the proof of Lemma 7 is immediate.

### C.1.3 PROOF OF LEMMA 7

*Proof.* To prove Lemma 7 simply note that the derivative of the norm squared of the gradient becomes negative:

$$\frac{d}{dt}\|\nabla \mathrm{Tr} D^2 \mathcal{L}(\theta(t))\|^2 = -\nabla \mathrm{Tr} D^2 \mathcal{L}(\theta(t))^\top \nabla^2 \mathrm{Tr} D^2 \mathcal{L}(\theta(t)) \nabla \mathrm{Tr} D^2 \mathcal{L}(\theta(t)) \leq 0.$$

$\square$

Next, we show a strict positive definite property for the hessian of approximate stationary points under our Assumptions 1, 2, and 3.

### C.2 STRONG LOCAL G-CONVEXITY

It turns out we can obtain an exponential decay on the norm of the gradient at approximate stationary points, given explicit positive lower bounds on $\phi'$ and $\phi'''$ stated in Assumption 1. As we mentioned, from Lemma 15 the gradient flow does not exit a bounded region around $\theta^*$ on $\mathcal{M}$, which means Assumption 1 automatically holds for some positive constants $\varrho_1, \varrho_2$ for the flow in Equation equation 3, given that we have the weaker condition $\phi', \phi''' > 0$ everywhere; this is because of compactnesss of bounded regions in the Euclidean space.

We state the exponential decay in Lemma 9.

**Lemma 9** (Decay of norm of gradient). *Under assumptions 1, 1, and 2, given time $t_0$ such that $\|\nabla \mathrm{Tr} D^2 \mathcal{L}(\theta(t_0))\| \leq \sqrt{\mu}\beta$, then the norm of gradient of $\mathrm{Tr} D^2 \mathcal{L}$ is exponentially decreasing for times $t \geq t_0$. Namely,*

$$\|\nabla \mathrm{Tr} D^2 \mathcal{L}(\theta(t))\|^2 \leq \|\nabla \mathrm{Tr} D^2 \mathcal{L}(\theta(t_0))\|^2 e^{-(t-t_0)\varrho_1 \varrho_2 \mu}.$$

The key idea to obtain this exponential decay in Lemma 9 is that at approximate stationary points $\theta$, under Assumption 1, the Hessian of $\mathrm{Tr} D^2 \mathcal{L}$ becomes strictly positive definite on a subspace of $\mathcal{T}_\theta(\mathcal{M})$ which is spanned by $\{x_i\}_{i=1}^n$ in the coordinates corresponding to $\theta_j$. Fortunately, the gradient $\nabla \mathrm{Tr} D^2 \mathcal{L}(\theta)$ is in this subspace, hence we can obtain the following Lemma. We show this formally in Lemma 10, proved in Appendix D.4.

**Lemma 10** (Strong convexity). *Under Assumptions 1, 1, and 2 we have for $\sqrt{\mu}\beta$-stationary $\theta$, i.e. when $\|\nabla_\theta \mathrm{Tr} D^2 \mathcal{L}\| \leq \sqrt{\mu}\beta$,*

$$\nabla \mathrm{Tr} D^2 \mathcal{L}(\theta)^\top \left(\nabla^2 \mathrm{Tr} D^2 \mathcal{L}(\theta)\right) \nabla \mathrm{Tr} D^2 \mathcal{L}(\theta) \geq \varrho_1 \varrho_2 \mu \|\nabla \mathrm{Tr} D^2 \mathcal{L}(\theta)\|^2.$$

### C.2.1 PROOF OF LEMMA 9

*Proof.* Using Lemma 10, we get positive definite property of the Hessian $\nabla^2 \mathrm{Tr} D^2 \mathcal{L}$ on the manifold by Lemma 10, which implies

$$\frac{d\|\nabla \mathrm{Tr} D^2 \mathcal{L}(\theta(t))\|^2}{dt} = -\nabla \mathrm{Tr} D^2 \mathcal{L}(\theta(t))^\top \nabla^2 \mathrm{Tr} D^2 \mathcal{L}(\theta(t)) \nabla g(\theta(t))$$
$$\leq -\varrho_1 \varrho_2 \mu \|\nabla \mathrm{Tr} D^2 \mathcal{L}(\theta(t))\|^2.$$

This inequality implies an exponential decay in the size of the gradient, namely for any $t \geq t_0$

$$\|\nabla \mathrm{Tr} D^2 \mathcal{L}(\theta(t))\|^2 \leq \|\nabla \mathrm{Tr} D^2 \mathcal{L}(\theta(t_0))\|^2 e^{-(t-t_0)\varrho_1 \varrho_2 \mu}.$$

$\square$

Next, we prove Theorem 3.

### C.3 Proof of Theorem 3

Using the non-decreasing property and exponential decay of the gradient at approximate stationary points that we showed in Lemmas 7 and 9 respectively, we bound the rate of convergence of the gradient flow equation 3 stated in Theorem 3. The idea is that once the flow enters the approximate stationary state, it gets trapped there by the local g-convexity. Finally, under Assumption 1, we show a semi-monotonicity property of $\mathrm{Tr}D^2\mathcal{L}$ in Lemma 6, proved in Appendix D.5; namely, having small gradient at a point on the manifold implies that the neurons are close to an optimal point $\theta^*$, as defined in Equation equation 4, in the subspace spanned by data. This automatically translates our rate of decay of gradient in Lemma 9 to a convergence rate of $\theta(t)$ to $\theta^*$.

First, note that once the norm of gradient satisfies

$$\|\nabla \mathrm{Tr}D^2\mathcal{L}(\theta(t))\| \leq \epsilon$$

for any $\epsilon \leq \sqrt{\mu}\beta$, it remains bounded by $\epsilon$ due to the non-increasing property of the norm of gradient guaranteed by Lemma 7. Therefore, if for some time $t$ we have $\|\nabla \mathrm{Tr}D^2\mathcal{L}(\theta(t))\| \leq \epsilon$ then $\|\nabla \mathrm{Tr}D^2\mathcal{L}(\theta(t'))\| \leq \epsilon$ for all $t' \geq t$. On the other hand, observe that the implicit regularizer decays with rate of squared norm of its gradient along the flow:

$$\frac{d\mathrm{Tr}D^2\mathcal{L}(\theta(t))}{dt} = -\|\nabla \mathrm{Tr}D^2\mathcal{L}(\theta(t))\|^2.$$

Therefore, if for some time $t$ we have $\|\nabla \mathrm{Tr}D^2\mathcal{L}(\theta(t))\| \geq \epsilon$ we can lower bound the value of $\mathrm{Tr}D^2\mathcal{L}(\theta)$ at time $t$ as

$$0 \leq \mathrm{Tr}D^2\mathcal{L}(\theta(t)) \leq \mathrm{Tr}D^2\mathcal{L}(\theta(0)) - \int_{t'=0}^{t} \|\nabla \mathrm{Tr}D^2\mathcal{L}(\theta(t'))\|^2 dt'$$

$$\leq \mathrm{Tr}D^2\mathcal{L}(\theta(0)) - \int_{t'=0}^{t} \epsilon^2 dt' = \mathrm{Tr}D^2\mathcal{L}(\theta(0)) - t\epsilon^2,$$

which implies

$$t \leq \mathrm{Tr}D^2\mathcal{L}(\theta(0))/\epsilon^2.$$

To further strengthen this into an exponential convergence rate, we know once the norm of the gradient goes below $\sqrt{\mu}\beta$ it will decay exponentially fast due to Lemma 9. From the first part, it takes at most $\mathrm{Tr}D^2\mathcal{L}(\theta(0))/(\mu\beta^2)$ time for the norm of gradient to go under threshold $\sqrt{\mu}\beta$, then after time $\log((\mu\beta^2/\epsilon'^2) \vee 1)/(\varrho_1\varrho_2\mu)$ it goes below $\epsilon'$ using the exponential decay. Hence, the overall time $t_{\epsilon'}$ it takes so that norm of the gradient goes below threshold $\epsilon'$ is bounded by

$$t_{\epsilon'} \leq \frac{\mathrm{Tr}D^2\mathcal{L}(\theta(0))}{\mu\beta^2} + \frac{\log((\mu\beta^2/\epsilon'^2) \vee 1)}{\varrho_1\varrho_2\mu}. \tag{16}$$

Now we want to combine this result with Lemma 6. For the convenience of the reader, we restate Lemma 6.

**Lemma** (see Lemma 6). *Suppose $\|\nabla \mathrm{Tr}D^2\mathcal{L}(\theta)\| \leq \delta$. Then, for all $i \in [n], j \in [m]$,*

$$\left| \theta_j^\top x_i - \theta_j^{*\top} x_i \right| \leq \delta/(\sqrt{\mu}\varrho_1\varrho_2),$$

*for a $\theta^*$ as defined in equation 4.*

Combining Equation equation 16 with Lemma 6, we find that to guarantee the property

$$|\theta^\top(t)_j x_i - \theta_j^{*\top} x_i| \leq \epsilon, \forall i \in [n], j \in [m], \tag{17}$$

we need to pick $\epsilon' = \epsilon\sqrt{\mu}\varrho_1\varrho_2$, which implies that we have condition equation 17 guaranteed for all times

$$t \geq \frac{\mathrm{Tr}D^2\mathcal{L}(\theta(0))}{\mu\beta^2} + \frac{\log(\beta^2/(\varrho_1^2\varrho_2^2\epsilon^2) \vee 1)}{\varrho_1\varrho_2\mu}.$$

# D    REMAINING PROOFS FOR CHARACTERIZING THE STATIONARY POINTS AND CONVERGENCE

## D.1    PROOF OF LEMMA 2

For any $\theta \in \mathbb{R}^{md}$ we can write

$$D^2\mathcal{L}(\theta) = 2\sum_{i=1}^{n} Df_i(\theta)Df_i(\theta)^\top + 2\sum_{i=1}^{n} D^2 f_i(\theta)(f_i(\theta) - y_i).$$

But from $\mathcal{L}(\theta) = 0$ we get for every $i \in [n]$, $f_i(\theta) = 0$, which implies

$$D^2\mathcal{L} = 2\sum_{i=1}^{n} Df_i(\theta)Df_i(\theta)^\top.$$

Taking trace from both sides completes the proof.

## D.2    PROOF OF LEMMA 3

Note that the gradient of the neural network function calculated on $x_i$ with respect to the parameter $\theta_j$ of the $j$th neuron is

$$D_{\theta_j} f_i(\theta) = \phi'(\theta_j^\top x_i) x_i.$$

This implies

$$\|Df_i(\theta)\|^2 = \sum_{j=1}^{m} \phi'(\theta_j^\top x_i)^2. \tag{18}$$

Summing Equations equation 18 for all $j \in [m]$ completes the proof.

## D.3    PROOF OF LEMMA 5

Combining Facts 1 and 2 implies

$$\begin{aligned}
\nabla^2 F(\theta)[u, w] &= \left\langle \nabla_u(\nabla F(\theta)), w \right\rangle \\
&= \left\langle P_\theta\Big(D(P_\theta(DF(\theta)))[u]\Big), w \right\rangle \\
&\overset{equation\ 19}{=} \left\langle D(P_\theta(DF(\theta)))[u], w \right\rangle \tag{19} \\
&= \left\langle D\Big((I - P_\theta^N)(DF(\theta))\Big)[u], w \right\rangle, \tag{20}
\end{aligned}$$

where Equality equation 19 follows because $w \in \mathcal{T}_\theta(\mathcal{M})$, the fact that for any vector $v$,

$$v = P_\theta(v) + P_\theta^N(v),$$

and that the part $P_\theta^N\Big(D(P_\theta(DF(\theta)))(u)\Big)$ has zero dot product with $w$. Now note that from Lemma 1, the rows of the Jacobian matrix $Df(\theta)$ spans the normal space $\mathcal{T}_\theta^N(\mathcal{M})$ for any $\theta \in \mathcal{M}$. Therefore, the projection matrix onto the normal space $\mathcal{T}_\theta^N(\mathcal{M})$ at point $\theta \in \mathcal{M}$ regarding the operator $P_\theta^N$ is given by

$$P_\theta^N(v) = Df(\theta)^\top (Df(\theta)Df(\theta)^\top)^{-1} Df(\theta)v. \tag{21}$$

Plugging Equation equation 21 into Equation equation 20, we get:

$$\begin{aligned}
\nabla^2 F(\theta)[u, w] &= \left\langle D\Big((I - Df(\theta)^\top (Df(\theta)Df(\theta)^\top)^{-1} Df(\theta))DF(\theta)\Big)[u], w \right\rangle \\
&= w^\top D\Big((I - Df^\top(DfDf^\top)^{-1}Df)DF\Big)[u] \\
&= w^\top (I - Df(\theta)^\top (Df(\theta)Df(\theta)^\top)^{-1} Df(\theta))D^2 F(\theta)u \\
&\quad + w^\top D\Big(I - Df(\theta)^\top (Df(\theta)Df(\theta)^\top)^{-1} Df(\theta)\Big)[u]DF(\theta), \tag{22}
\end{aligned}$$

where in the last line we used the chain rule. But note that $w \in \mathcal{T}_\theta(\mathcal{M})$, which implies $P_\theta(w) = w$. This means
$$(I - Df(\theta)^\top (Df(\theta)Df(\theta)^\top)^{-1}Df(\theta))w = w.$$
Plugging this back into Equation equation 22 and noting the fact that $D(I)[u] = 0$,

$$\nabla^2 F(\theta)[u, w] = w^\top D^2 F(\theta)u - w^\top D\Big(Df(\theta)^\top (Df(\theta)Df(\theta)^\top)^{-1}Df(\theta)\Big)[u]DF(\theta). \quad (23)$$

Now regarding the second term in Equation equation 23, note that the directional derivative in direction $u$ can either hit the $Df(\theta)$ terms or the middle part $(Df(\theta)Df(\theta)^\top)^{-1}$, i.e. we get

$$w^\top D\Big(Df(\theta)^\top (Df(\theta)Df(\theta)^\top)^{-1}Df(\theta)\Big)[u] = w^\top D^2 f(\theta)[u]^\top (Df(\theta)Df(\theta)^\top)^{-1}Df(\theta)$$
$$+ w^\top Df(\theta)^\top D\Big((Df(\theta)Df(\theta)^\top)^{-1}Df(\theta)\Big)[u]. \quad (24)$$

Now the key observation here is that because $w \in \mathcal{T}_\theta(\mathcal{M})$, we have
$$Df(\theta)w = 0,$$

which means the second term in Equation equation 24 is zero. Plugging this back into Equation equation 23 implies

$$\nabla^2 F(\theta)[u, w] = w^\top D^2 F(\theta)u - D^2 f(\theta)[u, w]^\top (Df(\theta)Df(\theta)^\top)^{-1}Df(\theta)DF(\theta),$$

which completes the proof.

### D.4 PROOF OF LEMMA 10

Let $\mathcal{N}$ be the subspace of $\mathcal{T}_\theta(\mathcal{M})$ which can be represented by

$$v = \Big[\sum_{i=1}^n \nu_i^j x_i\Big]_{j=1}^m, \quad (25)$$

for arbitrary coefficients $(\nu_i^j)_{i=1,\dots,n, j=1,\dots,m}$. First, we show that $\nabla \mathrm{Tr} D^2 \mathcal{L}(\theta) \in \mathcal{N}$. Recall the definition of the gradient of $\mathrm{Tr}D^2\mathcal{L}$ on $\mathcal{M}$, i.e.

$$\nabla \mathrm{Tr} D^2 \mathcal{L}(\theta) = P_\theta(D\mathrm{Tr}D^2\mathcal{L}(\theta)) = D\mathrm{Tr}D^2\mathcal{L}(\theta) - P_\theta^N(D\mathrm{Tr}D^2\mathcal{L}(\theta)).$$

Note that $P_\theta^N(D\mathrm{Tr}D^2\mathcal{L}(\theta)) \in \mathcal{N}$. This is because from Lemma 1 we know that $\{Df_i(\theta)\}_{i=1}^n$ is a basis for $\mathcal{T}_\theta^N(\mathcal{M})$ and each $Df_i(\theta)$ is clearly in the form equation 25. Now we argue for any vector $v \in \mathcal{N}$, we have

$$\nabla^2 \mathrm{Tr} D^2 \mathcal{L}(\theta)[v, v] \geq \mu \varrho_1 \varrho_2 \|v\|^2. \quad (26)$$

Let $v \triangleq \Big[\sum_{i=1}^n \nu_i^j x_i\Big]_{j=1}^m$. Note that for all $j \in [m]$, from Assumptions 1 and 2:

$$\nabla^2 \mathrm{Tr} D^2 \mathcal{L}(\theta)[v, v] = \sum_{j=1}^m \sum_{i=1}^n \phi'(\theta_j^\top x_i)\phi'''(\theta_j^\top x_i)\Big((\sum_{i'=1}^n \nu_{i'}^j x_{i'})^\top x_i\Big)^2$$
$$\geq \varrho_1 \varrho_2 \sum_{j=1}^m \sum_{i=1}^n \Big((\sum_{i'=1}^n \nu_{i'}^j x_{i'})^\top x_i\Big)^2$$
$$\geq \varrho_1 \varrho_2 \mu \sum_{j=1}^m \|\sum_{i'=1}^n \nu_{i'}^j x_{i'}\|^2,$$

where in the last inequality, we used the fact that from Assumption 2, for any vector $x$ in the subspace spanned by $\{x_i\}_{i=1}^n$ we have $x^\top X X^\top x \geq \mu \|x\|^2$. Combining this with Equation equation 15 in the proof of Lemma 4 completes the proof.

Next, we prove Lemma 6 which translates the approximate stationary property into topological closeness to a global optimum.

## D.5 PROOF OF LEMMA 6

Similar to the proof of Lemma 4, we have equation 13:

$$|\phi''(\theta_j^\top x_i) - \alpha_i'|\varrho_1 \leq |(\phi''(\theta_j^\top x_i) - \alpha_i')\phi'(\theta_j^\top x_i)| \leq \delta/(2\sqrt{\mu}),$$

which implies

$$|\phi''(\theta_j^\top x_i) - \alpha_i'| \leq \delta/(2\sqrt{\mu}\varrho_1). \tag{27}$$

Now since $\phi''' > 0$, we have that $\phi''$ is strictly monotone and invertible. Define

$$\nu_i' = \phi''^{-1}(\alpha_i'). \tag{28}$$

The fact that $\phi''' \geq \varrho_2$ provides us with a Lipschitz constant of $1/\varrho_2$ for $\phi''^{-1}$. Using this with equation 27 and equation 28 implies

$$\left|\theta_j^\top x_i - \nu_i'\right| \leq \delta/(2\sqrt{\mu}\varrho_1\varrho_2). \tag{29}$$

From equation 29 we want to show $|\nu_i - \nu_i'| \leq \delta/(2\sqrt{\mu}\varrho_1\varrho_2)$. Suppose this is not true. Then either $\nu_i > \nu_i' + \delta/(2\sqrt{\mu}\varrho_1\varrho_2)$ or $\nu_i < \nu_i' - \delta/(2\sqrt{\mu}\varrho_1\varrho_2)$. In the first case, using Equation equation 29 we get for all $j \in [m]$:

$$\nu_i > \theta_j^\top x_i,$$

which from the strict monotonicity of $\phi$ implies

$$y_i/m = \phi(\nu_i) > \phi(\theta_j^\top x_i),$$

which means $f_i(\theta) < y_i$. But this clearly contradicts with the fact that $\theta$ is on the manifold of zero loss, i.e. $f_i(\theta) = y_i$. In the other case $\nu_i < \nu_i' - \delta/(2\sqrt{\mu}\varrho_1\varrho_2)$ we can get a similar contradiction. Hence, overall we proved

$$|\nu_i - \nu_i'| \leq \delta/(2\sqrt{\mu}\varrho_1\varrho_2).$$

Combining this with Equation equation 29 completes the proof.

## E PROOF OF CONVERGENCE OF THE FLOW TO $\mathcal{M}$

In this section, we analyze the behavior of label noise SGD starting from $\theta_0$ with positive loss, when step size goes to zero. To this end, we study a different gradient flow than the one in Equation equation 3.

### E.1 GRADIENT FLOW REGARDING LABEL NOISE SGD IN THE LIMIT

In Lemma 11, we prove that under Assumptions 1, 2, the gradient flow with respect to the gradient $-D\mathcal{L}(\theta)$ converges to the manifold $\mathcal{M}$ exponentially fast. This is in particular important to in the proof of Theorem 1, in particular in proving that SGD with small enough step size will reach zero loss, because SGD in the limit of step size going to zero will converge to this gradient flow outside of $\mathcal{M}$. The key to show this result is a PL inequality that we prove in this setting for $\mathcal{L}$, in Lemma 12. Here we obtain explicit constants which is not necessary for proving Theorem 1.

**Lemma 11** (Convergence of the gradient flow to the manifold). *In the limit of step size going to zero, SGD initialized at a point $\theta_0$ converges to the following gradient flow*

$$\bar{\theta}(0) = \theta_0, \tag{30}$$

$$\bar{\theta}'(t) = -D\mathcal{L}(\bar{\theta}(t)). \tag{31}$$

*Under Assumption 1 and 2, after time*

$$t \geq \log(1/\epsilon)$$

*we have,*

$$\mathcal{L}(\bar{\theta}(t)) \leq e^{-Ct}\mathcal{L}(\bar{\theta}_0),$$

*for constant $C = 4m\mu\varrho_1^2$. Furthermore, $\theta(t)$ converges to $\tilde{\theta} \in \mathcal{M}$ such that for all $t > 0$*

$$\|\bar{\theta}(t) - \tilde{\theta}\| \leq \frac{2}{\sqrt{C}}e^{-Ct/2}\sqrt{\mathcal{L}(\bar{\theta}_0)}, \tag{32}$$

*and*

$$\mathrm{Tr}D^2\mathcal{L}(\tilde{\theta}) \leq \sum_{i=1}^n \sum_{j=1}^m \phi'^2(\theta_0^\top x_i) + \sum_{i=1}^n \max_{j=1}^m \{\phi'^2(\theta_{0j}^\top x_i) \pm \frac{2}{\sqrt{C}}\sqrt{\mathcal{L}(\bar{\theta}_0)}\}.$$

### E.2 PROOF OF LEMMA 11

First, taking derivative from $\mathcal{L}(\bar{\theta}(t))$ and using the PL inequality from Lemma 12:

$$\frac{d}{dt}\mathcal{L}(\bar{\theta}(t)) = -\|D\mathcal{L}(\bar{\theta})\|^2 \leq -4m\mu\varrho_1^2\mathcal{L}(\bar{\theta}),$$

which implies

$$\mathcal{L}(\bar{\theta}(t)) \leq e^{-4m\mu\varrho_1^2 t}\mathcal{L}(\theta_0). \tag{33}$$

Moreover, similar to page 38 in Li et al. (2021), given the PL constant $C = 4m\mu\varrho_1^2$ we have

$$\left\|\frac{d\bar{\theta}(t)}{dt}\right\| = \left\|\nabla\mathcal{L}(\bar{\theta}(t))\right\| \leq \frac{\left\|\nabla\mathcal{L}(\bar{\theta}(t))\right\|^2}{\sqrt{C\mathcal{L}(\bar{\theta}(t))}} = \frac{-\frac{d\mathcal{L}(\bar{\theta}(t))}{dt}}{\sqrt{C\mathcal{L}(\bar{\theta}(t))}} = -\frac{2}{\sqrt{C}}\frac{d\sqrt{\mathcal{L}(\bar{\theta})}}{dt}. \tag{34}$$

This implies

$$\|\bar{\theta}(t) - \bar{\theta}(0)\| \leq \int_0^t \left\|\frac{d\bar{\theta}(t)}{dt}\right\|dt \leq -\frac{2}{\sqrt{C}}\int_0^t \frac{d\sqrt{\mathcal{L}(\bar{\theta}(t))}}{dt}$$

$$= \frac{2}{\sqrt{C}}(\sqrt{\mathcal{L}(\bar{\theta}(0))} - \sqrt{\mathcal{L}(\bar{\theta}(t))}) \leq \frac{2}{\sqrt{C}}\sqrt{\mathcal{L}(\bar{\theta}(0))}. \tag{35}$$

Additionally, from Equation equation 34 we get for $t_1 \geq t_2$:

$$\|\bar{\theta}(t_1) - \bar{\theta}(t_2)\| \leq \frac{2}{\sqrt{C}}(\sqrt{\mathcal{L}(\bar{\theta}(t_1))}), \tag{36}$$

which combined with Equation equation 33 implies that $\theta(t)$ converges to some $\tilde{\theta}$. On the other hand, since $\mathcal{L}(\theta(t))$ converges to $\mathcal{L}(\tilde{\theta})$ by continuity of $\mathcal{L}$ and because $\mathcal{L}(\bar{\theta}(t))$ is going to zero, we conclude that the limit point $\tilde{\theta}$ should have zero loss, i.e. $\tilde{\theta} \in \mathcal{M}$. Now applying Equation equation 36 for $t_2 = t$ and sending $t_1$ to infinity implies for all $t > 0$:

$$\|\bar{\theta}(t) - \tilde{\theta}\| \leq \frac{2}{\sqrt{C}}\sqrt{\mathcal{L}(\bar{\theta}(t))} \leq \frac{2}{\sqrt{C}}e^{-Ct/2}\sqrt{\mathcal{L}(\theta_0)}.$$

On the other hand, recall that from Equation, trace of Hessian on the manifold is equal to

$$\mathrm{Tr}D^2\mathcal{L}(\theta) = \sum_{i=1}^n \left\|\nabla f_i(\bar{\theta})\right\|^2 = \sum_{i=1}^n\sum_{j=1}^m \phi'(\bar{\theta}_j^\top x_i)^2. \tag{37}$$

But under Assumption 1 we have that $\phi'^2$ is convex, because

$$\frac{d^2}{dz^2}\phi'(z)^2 = 2\phi''(z)^2 + 2\phi'(z)\phi'''(z) \geq 0.$$

But from Equation equation 35 we have

$$\sum_{j=1}^m \left|\bar{\theta}_j^\top(t)x_i - \bar{\theta}_j^\top(0)x_i\right| \leq \|\bar{\theta}(t) - \bar{\theta}(0)\| \leq \frac{2}{\sqrt{C}}\sqrt{\mathcal{L}(\bar{\theta}(0))}.$$

Combining the convexity and positivity of $\phi'^2$ with Equation equation 35 and the formula for $\left\|\nabla f_i(\theta)\right\|^2$:

$$\left|\left\|\nabla f_i(\bar{\theta}(t))\right\|^2 - \left\|\nabla f_i(\bar{\theta}(0))\right\|^2\right| \leq \max_{j=1}^m\{\phi'^2(\bar{\theta}_j^\top(0)x_i \pm \frac{2}{\sqrt{C}}\sqrt{\mathcal{L}(\bar{\theta}(0))}) - \phi'^2(\bar{\theta}_j^\top(0)x_i)\}$$

$$\leq \max_{j=1}^m\{\phi'^2(\theta_j^\top(0)x_i \pm \frac{2}{\sqrt{C}}\sqrt{\mathcal{L}(\bar{\theta}(0))})\}, \tag{38}$$

where $\pm$ above in the argument of $\phi'$ means we take maximum with respect to both $+$ and $-$. Summing Equation equation 38 for all $i$, we have for constant

$$C_2 := \sum_{i=1}^{n} \max_{j=1}^{m} \{\phi'^2(\bar{\theta}_j^\top(0)x_i \pm \frac{2}{\sqrt{C}}\sqrt{\mathcal{L}(\bar{\theta}(0))})\},$$

we have

$$\sum_{i=1}^{n} \left\|\nabla f_i(\bar{\theta}(t))\right\|^2 \leq \sum_{i=1}^{n} \left\|\nabla f_i(\theta_0)\right\|^2 + C_2.$$

Sending $t$ to infinity implies

$$\sum_{i=1}^{n} \left\|\nabla f_i(\tilde{\theta})\right\|^2 \leq \sum_{i=1}^{n} \left\|\nabla f_i(\theta_0)\right\|^2 + C_2.$$

But since $\tilde{\theta} \in \mathcal{M}$, from equation 37 we have

$$\mathrm{Tr}D^2\mathcal{L}(\tilde{\theta}) \leq \sum_{i=1}^{n}\sum_{j=1}^{m} \phi'(\theta_{0_j}^\top x_i)^2 + C_2,$$

which completes the proof.

### E.3 A PL INEQUALITY FOR $\mathcal{L}$

Next, in Lemma 12 we show a PL inequality for $\mathcal{L}$ under Assumptions 1 and 2, which we then use to show Lemma 11.

**Lemma 12** (PL inequality outside of manifold). *Under Assumptions 1 and 2, the loss $\mathcal{L}(\theta)$ satisfies a PL inequality of the form*

$$\|D\mathcal{L}(\bar{\theta})\|^2 \geq 4m\mu\varrho_1^2\mathcal{L}(\bar{\theta}).$$

*Proof.* Note that for all $j \in [m]$, $D_{\theta_j}\mathcal{L}(\bar{\theta})$ is given by

$$D_{\bar{\theta}_j}\mathcal{L}(\bar{\theta}) = 2\sum_{i=1}^{n}(r_{\bar{\theta},NN}(x_i) - y_i)\phi'(\bar{\theta}_j^\top x_i)x_i.$$

Hence

$$\|D_{\bar{\theta}_j}\mathcal{L}(\bar{\theta})\|^2 = 4\left\|\sum_{i=1}^{n}(r_{\bar{\theta},NN}(x_i) - y_i)\phi'(\bar{\theta}_j^\top x_i)x_i\right\|^2$$

$$\geq 4\mu\sum_{i=1}^{n}\phi'(\bar{\theta}_j^\top x_i)^2(r_{\bar{\theta},NN}(x_i) - y_i)^2$$

$$\geq 4\mu\varrho_1^2\mathcal{L}(\bar{\theta}). \tag{39}$$

summing Equation equation 39 for all neurons completes the proof. $\square$

## F PROOF OF THE FINAL GUARANTEE FOR SGD

In this section we prove Theorem 1.

### F.1 PROOF OF THEOREM 1

Our main theorem (Theorem 1) is a direct combination of Theorem 4.6 of Li et al. (2021), and Theorem 2 and Theorem 3. First note that from Lemma 11, the gradient flow in the limit of step size going to zero always converges to the manifold of zero, independent of the initialization. Therefore, the neighborhood $U$ in Theorem 4.6 of Li et al. (2021) is the whole $\mathbb{R}^{md}$, which implies that for

step size $\eta < \eta_0$ less than a certain threshold $\eta_0$, $\theta_{\lceil t/\eta^2 \rceil}$ is close enough to $\theta(t)$ in distribution, where $\theta(t)$ is the Riemannian gradient flow defined in Theorem 3 which is initialized as $\theta(0) = \tilde{\theta}$. Recall that $\tilde{\theta}$ is the limit point of the first gradient flow (before reaching the manifold $\mathcal{M}$) defined in Lemma 11.

Moreover, again from Lemma 11, the gradient flow $\bar{\theta}(t)$ defined in that Lemma remains within distance of $O(\frac{2}{\sqrt{C}}\sqrt{\mathcal{L}(\bar{\theta}_0)})$ of initialization. Therefore, trace of Hessian of the loss at $\tilde{\theta}$ is bounded and only depends on the initialization $\theta_0$. This enables us to have a bounded initial value for the second ODE in equation 3, depending only on the primary initialization $\bar{\theta}(0)$ (which can be outside the manifold). In particular, note that from Theorems 2 and 3, this flow converges to $\epsilon'$-proximity of a global minimizer $\theta^*$ of trace of Hessian on $\mathcal{M}$ in the sense of equation 5 after time at most $\tilde{O}(\log(1/\epsilon))$.

Now using the unit norm assumption $\|x_i\| = 1$ and picking the step size threshold $\eta_0$ small enough, we see that after at most $K = \Theta((1 + \log(1/\epsilon'))/\eta^2)$ iterations of label noise SGD for $\eta \leq \eta_0$, for all $j \in [m]$ and $i \in [n]$ we have with high probability

$$\left| \theta_{T_j}^\top x_i - \theta^*{}_j^\top x_i \right| \leq 2\epsilon'. \tag{40}$$

Therefore, picking $\epsilon' \leq \epsilon/2$, we get for all $i \in [n]$ and $j \in [m]$:

$$\left| \theta_{T_j}^\top x_i - \phi^{-1}(y_i/m) \right| \leq \epsilon.$$

Moreover, note that from Equation equation 32 and Lemma 15, $\theta$ remains in a ball of bounded radius depending only on $\theta_0$, hence $\mathcal{L}$ and $\mathrm{Tr} D^2 \mathcal{L}$ both have a bounded Lipschitz constant only depending on the initialization as well. Therefore, we pick $\epsilon'$ small enough so that we get $\mathcal{L}(\theta_K) \leq \epsilon$ and $\left| \mathrm{Tr} D^2 \mathcal{L}(\theta_K) - \mathrm{Tr} D^2 \mathcal{L}(\theta^*) \right| \leq \epsilon$ for the final iteration $K$ of label noise SGD. This completes the proof.

## G  DERIVATIVE FORMULAS

### G.1  DERIVATIVE TENSORS

In Lemma 13 we calculate the Jacobian of $f$ and the Hessian of $f_i$'s.

**Lemma 13.** *The Jacobian of $f$ is given by:*

$$Df = \begin{bmatrix} \phi'(\theta_1^\top x_1)x_1^\top & \cdots & \cdots \\ \cdots & \phi'(\theta_j^\top x_i)x_i^\top, \ \phi'(\theta_{j+1}^\top x_i)x_i^\top & \cdots \\ \cdots & \cdots & \cdots . \end{bmatrix}$$

*Moreover, the Hessian of $f_i$'s is given by*

$$D^2 f_i(.,.) = \begin{bmatrix} \phi''(x_i^\top \theta_1)x_i x_i^\top & 0 & \cdots \\ 0\cdots & \cdots & \\ 0\cdots & \phi''(x_i^\top \theta_j)x_i x_i^\top & 0\cdots \\ & \cdots & . \end{bmatrix}$$

*As a result*

$$D^2 f_i(u, u) = \sum_{j=1}^{N} \phi''(x_i^\top \theta_j)(x_i^\top u_j)2.$$

In this Lemma 14, we calculate the Hessian of the trace of Hessian regularizer in our two-layer setting.

**Lemma 14.** *For the implicit regularizer $F \triangleq \mathrm{Tr} D^2 \mathcal{L}$ regarding our two-layer network with mean squared loss as in equation 2, we have*

$$F(\theta) = \sum_{i=1}^{n} \sum_{j=1}^{m} \phi'^2(\theta_j^\top x_i)\|x_i\|^2 = \sum_i \sum_j \phi'^2(\theta_j^\top x_i),$$

*and*

$$D^2F(.,.) \tag{41}$$

$$= \sum_i \begin{bmatrix} (2\phi''^2(x_i^\top\theta_1) + \phi'''(x_i^\top\theta_1)\phi'(x_i^\top\theta_1))x_i x_i^\top & 0 & \cdots \\ 0\cdots & \cdots 0 \cdots \quad (2\phi''^2(x_i^\top\theta_j) + \phi'''(x_i^\top\theta_j)\phi'(x_i^\top\theta_j))x_i x_i^\top & 0\cdots \\ \cdots & . \end{bmatrix} \tag{42}$$

*Hence*

$$D^2F(\theta)[u,u] = \sum_{i=1}^n \sum_{j=1}^m (2\phi''^2(x_i^\top\theta_j) + 2\phi'''(x_i^\top\theta_j)\phi'(x_i^\top\theta_j))\langle x_i, u_j\rangle^2. \tag{43}$$

## H    OTHER PROOFS

In Lemma 15 we show that the gradient flow in equation 3 remains in a bounded region.

**Lemma 15** (Bounded region). *Under the positivity $\phi''' > 0$, the gradient flow $\frac{d}{dt}\theta(t) = -\nabla\mathrm{Tr}D^2\mathcal{L}(\theta(t))$ remains in a bounded region for all time $t$.*

*Proof.* Note that the value of $\mathrm{Tr}D^2\mathcal{L}$ is decreasing along the flow, so for all $t \geq 0$:

$$\mathrm{Tr}D^2\mathcal{L}(\theta(t)) \leq \mathrm{Tr}D^2\mathcal{L}(\theta_0).$$

But noting the formula of $\mathrm{Tr}D^2\mathcal{L}$ in Lemma 3, we get for all $j \in [m]$ and $i \in [n]$:

$$\phi'(\theta_j^\top x_i)^2 \leq \mathrm{Tr}D^2\mathcal{L}(\theta_0). \tag{44}$$

Now let $z^*$ be the minimizer of $\phi'$. Now from the assumption of the Lemma we have $\phi'''(z^*) > 0$, which implies from continuity of $\phi'''$ that for $\epsilon', \delta > 0$ we have for all $z \in (z^* - \epsilon', z^* + \epsilon')$,

$$\phi'''(z) \geq \delta'.$$

This means for all $z \leq z^* - \epsilon'$ we have $\phi''(z) \leq -\epsilon'^2\delta'/2$ and for $z \geq z^* + \epsilon'$ we have $\phi'(z) \geq \epsilon'^2\delta'/2$. Even more, for any $|z - z^*| \geq \epsilon'$ we have

$$\phi'(z) \geq (|z - z^*| - \epsilon')\epsilon'\delta' + \epsilon'^2\delta'/2 = (|z - z^*| - \epsilon'/2)\epsilon'\delta'.$$

Therefore, equation 44 implies for $z = \theta_j^\top x_i$:

$$(|z - z^*| - \epsilon'/2)\epsilon'\delta' \leq \mathrm{Tr}D^2\mathcal{L}(\theta_0),$$

or

$$|z - z^*| \leq \mathrm{Tr}D^2\mathcal{L}(\theta_0)/(\epsilon'\delta') + \epsilon'/2.$$

This implies the boundedness of the arguemts of all the activations for all data points, which completes the proof. $\square$

**Lemma 16** ($\mathcal{M}$ is well-defined). *The intersection of the sub-level sets $f_i(\theta) = y_i$, defined in Section 4 as $\mathcal{M}$, is well-defined as a differentiable manifold.*

*Proof.* Due to a classical application of the implicit function theorem after a linear change of coordinates using a basis for the kernel of the Jacobian, it turns out that there exists a local chart for $\mathcal{M}$ around each point $\theta \in \mathcal{M}$ for which the Jacobian matrix is non-degenerate. But due to Lemma 17 the Jacobian is always non-degenerate, so $\mathcal{M}$ is indeed well-defined as a manifold. $\square$

**Lemma 17** (Non-singularity of the Jacobian). *Under assumption 1, the Jacobian $Df$ is non-singular, i.e. its rank is equal to the number of its rows.*

*Proof.* For an arbitrary data point $\theta_j$, we show the non-singularity of the submatrix of $Df$ whose columns corresponds to $\theta_j$, namely $D_{\theta_j} f$. Any linear combination of the rows of this matrix is of the form

$$\sum_{i=1}^{n} \alpha_i \phi'(\theta_j^\top x_i) x_i. \tag{45}$$

But from the coherence assumption in 1, we see that $\{x_i\}_{i=1}^n$ are linearly independent, so the linear combination in equation 45 is also non-zero, which implies the non-singularity of $D_{\theta_i} f$ and completes the proof. □

**Lemma 18** (Cube activation). *If the distribution of the labels does not have a point mass on zero (i.e. if $\mathbb{P}(y = 0) = 0$), then for the cube activation the manifold $\mathcal{M}$ is well-defined, and Theorem 2 also holds.*

*Proof.* To show that $\mathcal{M}$ is well-defined, similar to Lemma 16 we show the jacobian $Df(\theta)$ is non-degenerate for all $\theta$. To this end, it is enough to show that for each $x_i$, there is a $j$ such that $D_{\theta_j} f_i$ is non zero. This is because $D_{\theta_j} f_i$ is always a scaling of $x_i$, so if it is non-zero, then it is a non-zero scaling of $x_i$, and since $x_i$'s are linearly independent from Assumption 1 the non-degeneracy of $Df$ follows. To show the aforementioned claim, note that for each $\theta \in \mathcal{M}$, from the assumption on the distribution of the label, we have with probability one:

$$\sum_{j=1}^{m} \phi(\theta_j^\top x_i) = y_i \neq 0.$$

Therefore, there exists $\tilde{j}$ such that

$$\phi(\theta_j^\top x_i) \neq 0,$$

which implies

$$\theta_j^\top x_i \neq 0,$$

which means

$$\phi'(\theta_j^\top x_i) \neq 0.$$

Therefore,

$$D_{\theta_j} f_i(\theta) = \phi'(\theta_j^\top x_i) x_i \neq 0,$$

and the proof of the non-degeneracy of $Df$ is complete.

Next, we show that proof of Theorem 2 is still valid even though Assumption 1 does not hold for $\phi(x) = x^3$. This is because the only argument that we use in the proof of Theorem 2 which depends on Assumption 1 is the invertibility of $\phi$ and $\phi''$ which holds for the cube activation. Hence, the proof of complete.

□

## I  G-CONVEXITY

In the following Lemma 19, we show that PSD property of the Hessian of an arbitrary smooth function $F$ on $\mathcal{M}$ at point $\theta$ implies that its locally g-convex at $\theta$.

**Lemma 19** (Local g-convexity). *For a function $F$ on manifold $\mathcal{M}$, given that its Hessian is PSD in an open neighborhood $\mathcal{C} \subseteq \mathcal{M}$, then $F$ is g-convex in $\mathcal{C}$ on $\mathcal{M}$.*

*Proof.* Consider a geodesic $\gamma(t) \subseteq \mathcal{M}$. Then taking derivative of $F$ on $\gamma$ with respect to $t$:

$$\frac{d}{dt} F(\gamma(t)) = \langle \nabla F(\gamma(t)), \gamma'(t) \rangle,$$

and for the second derivative

$$\frac{d^2}{dt^2}F(\gamma(t)) = \langle \nabla F(\gamma(t)), \nabla_{\gamma'(t)}\gamma'(t)\rangle + \langle \nabla^2 F(\gamma(t))\gamma'(t), \gamma'(t)\rangle$$
$$= \langle \nabla F(\gamma(t)), \nabla_{\gamma'(t)}\gamma'(t)\rangle,$$

where we used the fact that $\nabla_{\gamma'(t)}\gamma'(t) = 0$ for a geodesic, and the fact that the Hessian of $F$ is PSD. Therefore, $F$ is convex in the normal sense on any geodesic in $\mathcal{C}$, which means it is g-convex in $\mathcal{C}$. $\qquad\square$

## J    BACKGROUND ON RIEMANNIAN GEOMETRY

In this section, we recall some of basic concepts in differential geometry, related to our analysis in this work. The material here is directly borrowed from Gatmiry & Vempala (2022); Gatmiry et al. (2023). For a more detailed treatment on Differential Geometry, we refer the reader to Do Carmo (2016).

**Abstract manifold.**    A manifold $\mathcal{M}$ is a topological space such that for an integer $n \in \mathbb{N}$ and given each point $p \in \mathcal{M}$, there exists an open set $U$ around $p$ such that $U$ is a homeomorphism to an open set of $\mathbb{R}^n$. We call a function $f$ defined on $\mathcal{M}$ smooth if for every $p \in \mathcal{M}$ and open set $U$ around $p$ which is homeomorphism to an open set $V \subseteq \mathbb{R}^n$ with mapping $\phi$, then $f \circ \phi$ is a smooth function of $V$. This is how we define vector fields on $\mathcal{M}$ later on.

Note that the manifold we consider in this work is a sub-manifold embedded in the Euclidean space, given by the intersection of sublevel sets of $f_i(\theta)$'s. Given that the Jacobian of $f = (f_i)_{i=1}^n$ is non-singular(Lemma 17), the manifold $\mathcal{M} = \{x \mid \forall i \in [n]: f_i(x) = y_i\}$ is well-defined.

**Tangent space.**    For any point $p \in \mathcal{M}$, one can define the notion of tangent space for $p$, $T_p(\mathcal{M})$, as the set of equivalence class of differentiable curves $\gamma$ starting from $p$ ($\gamma(0) = p$), where we define two such curves $\gamma_0$ and $\gamma_1$ to be equivalent if for any function $f$ on the manifold:

$$\frac{d}{dt}f(\gamma_0(t))\big|_{t=0} = \frac{d}{dt}f(\gamma_1(t))\big|_{t=0}.$$

When the manifold is embedded in $\mathbb{R}^d$, the tangent space at $p$ can be identified by the subspace of $\mathbb{R}^d$ obtained from the tangent vectors to curves on $\mathcal{M}$ passing through $p$.

**Vector field.**    A vector field $V$ is a smooth choice of a vector $V(p) \in T_p(\mathcal{M})$ in the tangent space for all $p \in \mathcal{M}$.

**Metric and inner product.**    A metric $\langle.,.\rangle$ is a tensor on the manifold $\mathcal{M}$ that is simply a smooth choice of a symmetric bilinear map on the tangent space of each point $p \in \mathcal{M}$. Namely, for all $w, v, z \in T_p(\mathcal{M})$:

$$\langle v + w, z\rangle = \langle v, z\rangle + \langle w, z\rangle,$$
$$\langle \alpha v, \beta w\rangle = \alpha\beta\langle v, w\rangle.$$

**Covariant derivative.**    Given two vector fields $V$ and $W$, the covariant derivative, also called the Levi-Civita connection $\nabla_V W$ is a bilinear operator with the following properties:

$$\nabla_{\alpha_1 V_1 + \alpha_2 V_2}W = \alpha_1\nabla_{V_1}W + \alpha_2\nabla_{V_2}W,$$
$$\nabla_V(W_1 + W_2) = \nabla_V(W_1) + \nabla_V(W_2),$$
$$\nabla_V(\alpha W_1) = \alpha\nabla_V(W_1) + V(\alpha)W_1$$

where $V(\alpha)$ is the action of vector field $V$ on scalar function $\alpha$. Importantly, the property that differentiates the covariant derivative from other kinds of derivaties over the manifold is that the covariant derivative of the metric is zero, i.e., $\nabla_V g = 0$ for any vector field $V$. In other words, we have the following intuitive rule:

$$\nabla_V\langle W_1, W_2\rangle = \langle \nabla_V W_1, W_2\rangle + \langle W_1, \nabla_V W_2\rangle.$$

Moreover, the covariant derivative has the property of being torsion free, meaning that for vector fields $W_1, W_2$:

$$\nabla_{W_1} W_2 - \nabla_{W_2} W_1 = [W_1, W_2],$$

where $[W_1, W_2]$ is the Lie bracket of $W_1, W_2$ defined as the unique vector field that satisfies

$$[W_1, W_2]f = W_1(W_2(f)) - W_2(W_1(f))$$

for every smooth function $f$.

In a local chart with variable $x$, if one represent $V = \sum V^i \partial x_i$, where $\partial x_i$ are the basis vector fields, and $W = \sum W^i \partial x_i$, the covariant derivative is given by

$$\begin{aligned}
\nabla_V W &= \sum_i V^i \nabla_i W = \sum_i V^i \sum_j \nabla_i (W^j \partial x_j) \\
&= \sum_i V^i \sum_j \partial_i(W^j)\partial x_j + \sum_i V^i \sum_j W^j \nabla_i \partial x_j \\
&= \sum_j V(W^j)\partial x_j + \sum_i \sum_j V^i W^j \sum_k \Gamma_{ij}^k \partial x_k = \\
&= \sum_k \big(V(W^k) + \sum_i \sum_j V^i W^j \Gamma_{ij}^k\big)\partial x_k.
\end{aligned}$$

The Christoffel symbols $\Gamma_{ij}^k$ are the representations of the Levi-Cevita derivatives of the basis $\{\partial x_i\}$:

$$\nabla_{\partial x_j} \partial x_i = \sum_k \Gamma_{ij}^k \partial x_k$$

and are given by the following formula:

$$\Gamma_{ij}^k = \frac{1}{2} \sum_m g^{km}(\partial_j g_{mi} + \partial_i g_{mj} - \partial_m g_{ij}).$$

Above, $g^{ij}$ refers to the $(i, j)$ entry of the inverse of the metric.

**Gradient.** For a function $F$ over the manifold, it naturally acts linearly over the space of vector fields: given a vector field $V$, the mapping $V(F)$ is linear in $V$. Therefore, by Riesz representation theorem there is a vector field $W$ where

$$V(F) = \langle W, V \rangle$$

for any vector field $V$. This vector field $W$ is defined as the gradient of $F$, which we denote here by $\nabla F$. To explicitly derive the form of gradient in a local chart, note that

$$V(F) = (\sum_i \partial x_i V_i)(F) = \sum_i V_i \partial_i F = \sum_{j,m,i} V_j g_{jm} g^{mi} \partial_i F = \langle V, \big(\sum_i g^{mi} \partial_i F\big)_m \rangle.$$

The above calculation implies

$$\nabla F = \sum_m (\sum g^{mi} \partial x_i F)\partial x_m.$$

So $g^{-1}DF$ is the representation of $\nabla F$ in the Euclidean chart with basis $\{\partial x_i\}$.

Given a smooth function $F$ on $\mathbb{R}^{md}$ and sub-manifold $\mathcal{M} \subseteq \mathbb{R}^{md}$, the gradient $\nabla F$ on $\mathcal{M}$ at any point $\theta \in \mathcal{M}$ is given by the Euclidean projection of the normal Euclidean gradient of $F$ at $\theta$ onto the tangent space $\mathcal{T}_\theta(\mathcal{M})$.

