# OpenReview forum: "Simplicity Bias of SGD via Sharpness Minimization"
_ICLR.cc/2024/Conference — Submitted to ICLR 2024_

### Official Review · Reviewer_dCJC · 2023-10-18

**Soundness:** 3 good
**Presentation:** 2 fair
**Contribution:** 3 good
**Rating:** 5
**Confidence:** 3

**Summary:**

The authors build on the work of Li, Wang, and Arora in ICLR 2022 (which they reference by means of its version in arXiv 2110.06914), in particular its result that SGD with label noise and infinitesimal learning rate traverses the manifold of zero loss in parameter space as the gradient flow that minimises the trace of the Hessian of the loss.  They investigate theoretically this gradient flow further for two-layer networks in which the second layer is fixed, and the derivative of the activation function is strictly positive and strictly convex; and for data points which are linearly independent and hence of cardinality not greater than their dimension.  There are two main results: first, that the first-order and global optima of the trace of the Hessian of the loss on the zero-loss manifold coincide, and are such that the projection of every first-layer weight vector in every such optimum onto the subspace spanned by the data points is a unique vector determined by the data, the network width, and the invertible activation function; and second, that gradient flow on the zero-loss manifold converges to such an optimum exponentially fast.  The theoretical results are supplemented by numerical experiments that explore weakening the assumption of data to allow cardinality greater than the dimension.

**Strengths:**

The theoretical results are proved in the appendix, which also contains helpful background material on manifold, tangent spaces, vector fields, covariant derivatives, etc.

The technique used to prove the convergence result is non-trivial, and the insight that it yields is potentially of wider interest: that the gradient on the zero-loss manifold being small implies that the trace of the Hessian of the loss has a positive semi-definite Hessian on the manifold, or equivalently it is locally geodesically convex.

**Weaknesses:**

The most involved part of the paper, Section 4.3, is hard to read.  The proof sketch for the first claim of Theorem 3 is split in two, with the proof sketch for the second claim being in the middle.  The functions $g$ and $\gamma$ do not seem to be defined in the main.

The paper contains various typos, LaTeX issues, English issues, and small errors: Assumption 2 is referred to as Assumption 1, the font of "NN" in $r_{\theta, \mathsf{NN}}$ is not consistent, the sentence "The novelty of our approach..." on page 3 does not completely parse, the definition of $\mathcal{M}$ in Section 4.1 should be in terms of the labels $y_i$, etc.  Those are some examples, I recommend to the authors to proof-read the whole paper.

Apparently no code is submitted as a supplement to the paper, which makes reproducing the numerical experiments harder.

**Questions:**

The assumptions on the activation function (Assumptions 1 and 3) are relatively strong.  Where exactly are they needed in this work, and what would be the obstacles to considering e.g. the ReLU activation?

It seems to me that the projections of the first-layer weight vectors onto the subspace orthogonal to the data points stay fixed throughout the training?  If that is the case, then this orthogonal subspace basically plays no role in the paper, so why not simply assume that $n = d$?  That would also make the optima in parameter space (in Theorem 2) unique?

What would constitute interesting future work?  The conclusion (Section 5) does not seem to contain any suggestions.

How related is the paper "Sharpness-Aware Minimization Leads to Low-Rank Features" by Maksym Andriushchenko, Dara Bahri, Hossein Mobahi, and Nicolas Flammarion in arXiv 2305.16292?

---

> ### Author Response · Authors · 2023-11-20
> **Response to reviewer dCJC**
>
> **The proof of section 4.3 is not well-written.**
>
> Thank you for the feedback, indeed we shifted the proof overview of the second part of Theorem 3 after the overview of the first part.
>
> **$g$ and $\gamma$ are not defined.**
>
> Thank you for spotting this, the single line that includes the notation $g,\gamma$ is indeed a typo based on our previous notation. We have corrected this in the new version.
>
> **Typos/No code provided.**
>
> We apologize for the typos, we have fixed them in the new version. We have also submitted our code.
>
>
> **Q1: Where are the assumptions on the activations used?**
>
> The first assumption is crucial in obtaining a closed-form for the stationary point of the trace of Hessian, hence it is used in the proof of Theorem 2. The second assumption (Assumption 3) arises naturally in showing the local g-convexity property of trace of Hessian at approximate stationary points, which we crucially use to obtain our convergence rate.
>
> **Q2: Why not consider $d=n$?**
>
> The point mentioned by the reviewer is indeed correct, the subspace orthogonal to the data points plays no role and does not change during the optimization, hence the proof is similar to the case $d=n$. We preferred to present our result in this slightly more general setting because in applications of neural networks with high-dimensional data, it is often the case that $d > n$. Note that the subspace of the data span, even though it has a lower dimension than the ambient space, is still informative about the ultimate function that we are trying to learn.
>
>
> **Q3: No future directions provided.**
>
> Thank you for your feedback, we added two ideas in the conclusion section about future directions that we think are interesting regarding our work.
>
> **Q4: How is this work related to "Sharpness-Aware Minimization Leads to Low-Rank Features"**
>
> In the paper by writing down the update rule of SAM for a relu network, the authors unravel the local tendency of SAM to make the features sparser at each step. They further confirm this hypothesis empirically for deep relu networks and see that this sparsity also leads to low-rank features. In comparison, our work rigorously proves the low-rank property of the learned features by label noise SGD for a class of two-layer networks. Notably, our result can be generalized to 1-SAM as well by applying the results in Wen et al., 2023.
>
> Given our changes and replies, we kindly ask the reviewer to increase her score.

---

> > ### Comment · Reviewer_dCJC · 2023-11-21
> >
> > Thank you for these responses.  I may adjust my score, depending also on the discussions with the other reviewers.

---

### Official Review · Reviewer_wfZ7 · 2023-10-30

**Soundness:** 4 excellent
**Presentation:** 4 excellent
**Contribution:** 3 good
**Rating:** 8
**Confidence:** 3

**Summary:**

This theory paper studies the relationship between flatness and simplicity.   Prior works have shown that label noise SGD and 1-SAM have an implicit bias towards flat minimizers (where flatness is quantified using trace of Hessian), but have not engaged with the question of what small Hessian trace implies about the network itself.  This paper has basically two main contributions.   First, they prove that for two-layer networks in a high-dimensional setting, under a certain assumption on the activation function, all of the flattest global minimizers admit a rank-1 feature matrix on the training dataset.  That is, all of the neurons are effectively identical as far as the training set is concerned.  This is a strong form of simplicity.  Second, they characterize the rate of convergence of label noise SGD to this set of flattest global minimizers.

**Strengths:**

It's an important open problem to understand the relationship between flatness and complexity.  This paper takes what I think is a good step in that direction by proving that, in a certain setting, all of the flattest global minimizers yield a rank-1 feature matrix.

**Weaknesses:**

The main weaknesses of the paper are the unrealistic assumptions:

1.  The paper assumes that the data dimension is larger than the number of data points, which means that a linear regressor could fit the dataset perfectly.  This is an unrealistic assumption and it presumably makes a lot of the theory easier (e.g. I assume this assumption is what enables the authors to prove the global convergence of label noise SGD for narrow nets on arbitrary data from arbitrary initializations; barring this assumption on the data dimension, global convergence is ordinarily not provable even for vanilla gradient descent, let alone label noise SGD, right?).  On the positive side, the authors show _empirically_ that a version of the simplicity bias does seem to exist even for the practical low-dimensional setting.  I think it would be very interesting to characterize this simplicity bias in more detail, even if it cannot be accompanied by a global convergence proof.

2.  The paper assumes that the third derivative of the activation function is strictly positive, which I believe rules out most real activation functions.  It would be interesting if the authors could discuss what kind of results could be established even in the absence of this strong assumption on the activation function.  From skimming the proofs, it seems that this assumption is needed in order to invert the second derivative of the activation function, but could we say anything interesting about an activation like tanh where the second derivative is not completely invertible but is invertible up to sign?

**Questions:**

- As discussed above, how far could you get without making the assumptions about (1) high data dimension and (2) third derivative positivity?  Let's say we didn't care about proving global convergence, or global convergence rate, and just cared about understanding the structure of the flattest global minimizers.

- The conclusion says: "we show that by initializing the neurons in the subspace of high-dimensional input data, all of the neurons converge into a single vector."  But, I can't find any part of the text that discusses either this special initialization, or the result that the neurons converge to a single vector.  By contrast, the main text proves that for _any_ initialization, all the neurons _agree entirely on the training dataset_, which is weaker than saying that all neurons are identical.  I think I do understand why the sentence in the conclusion follows from this: if the neurons are initialized within the span of the input data, then since they always move within the span of the input data, they must end up within the span of the input data; and there is only one possible set of weights that both lies within the span of the input data and matches the necessary targets, and this is a pseudoinverse.  But the paper did not discuss this explicitly.

  - If you added weight decay, would it be true that for any initialization, all the neurons converge to a single vector?

---

> ### Author Response · Authors · 2023-11-20
> **Response to reviewer wfZ7**
>
> **Q2 Equality of the neurons in the span of data.**
>
> Thank you for this comment, indeed we obtain the equality of the neurons in the span of data by the fact that their dot product with the data points are all equal.
>
>
> **Q3 What if we add Weight decay?**
>
> Adding tiny weight decay might work ($\lambda\ll \eta$) if we also train for a very long time ( $1/\eta\lambda$ steps), but we don't have a formal proof. Adding constant or moderate Weight decay might prevent training loss from going to 0 or hurt sharpness minimization.

---

### Official Review · Reviewer_xXHB · 2023-10-30

**Soundness:** 2 fair
**Presentation:** 2 fair
**Contribution:** 2 fair
**Rating:** 5
**Confidence:** 4

**Summary:**

The paper under study presents results on the link between sharpness minimization and the simplicity bias that has been observed many times throughout neural networks training tasks.

More precisely, they study the result of the Trace of Hessian minimization program proposed by Li et al. (2021), on a problem parametrized by a one hidden layer neural network with fixed to 1 outer layer.
They show that such a simplicity bias happens.

**Strengths:**

A strength  of the paper is that the result is clear: for the specific prediction function, the asymptotic flow described in Li et al. (2021), shows a simplicity bias and converges.

**Weaknesses:**

The results and the way the paper is articulated lacks clearness and some parts are largely overclaimed. Here is a list of potential improvement regarding these:

- First, *it has to be clear that the authors consider* the Li et al. asymptotic flow for granted. They study it in a particular setting of the neural network with frozen outer layer and specific activation. Let me reming the authors that the result of Li et al is only asymptotic and the regime of label noise + step size + time scale that is described in this is not general at all. In other words, they should clarify this explicitly when stating their result (eg theorem 1).

- Second, the authors claim that *for the ease of the exposition*, they set the second layer to one. If there is not clear proof of the fact that the result can be adapted from this, I do not believe it it an easy step to adapt their result to such a setup. Even though I do not expect things to change qualitatively.

- Third, the Assumptions 1 and 3 on the activation function seems very restrtictive: relu, sigmoid or tanh, which are the popular ones do not seem to be covered.

- Finally, even though the Theorem 1 is informative, there is a need for more precision at this stage. To be concrete, there is  no clear definition of label noise SGD, no quantitative dependency on the step size and the noise of SGD… and for a fact: Li et al. result is asymptotic and non quantitative in essence.  Any quantitative and non-asymptotic result on this program is significantly difficult to obtain, see e.g.  **Label noise (stochastic) gradient descent implicitly solves the Lasso for quadratic parametrisation**, L. Pillaud-Vivien, et al. COLT 2022 on a specific prediction function.

On the simplicity bias and the label noise structure the authors might want to look at the following paper: **Sgd with large step sizes learns sparse features**. M Andriushchenko, et al., ICML 2023.

**Questions:**

See above for improvements.

Here is a list of typos:

- page 2: linearly converges
- Page 2: manifolds of minimisers
- Page 3 the paragraph with ‘the novelty of our approach’ has many typos and is not clear. Try to use ‘the’,
    - we characterise the convergence on the manifold…
    - Far com the stationary points of the Hessian trace optimization problem
    - We show that this implies
    - There is no convexity —> convexity of what ?
- Page 4:
        - twice equation equation 1,
        - non=degenerecy
- Page 5:
        - assumptions 1 and 1
        - Assumption 1 and 2 should be 1 and 3.
- Page 6 : problem with assumptions again

---

> ### Author Response · Authors · 2023-11-20
> **Response to reviewer xXHB**
>
> **Q1: The result of Li et al is asymptotic and not general.**
>
> Response: We want to clarify that we have already clearly stated that our main theorem (Theorem 1) only holds for a sufficiently small learning rate, label noise SGD, and that with step size $\eta$, the simplicity bias happens in $\tilde O(1/\eta^2)$ iterations. Following the reviewer's suggestion, we also stressed in the main text before Theorem 1, that our theorem " holds for sufficiently small learning rate."
>
> **Q2: The proof of the generalization to unequal second-layer weights is not clear.**
>
> Our analysis indeed extends to the case when the second layer weights are not all equal, as discussed in the proof of Theorem 2 (page 7 the paragraph "This characterizes...".) In such case, our simplicity bias will be slightly different, in the sense that the $\phi''$ image of the feature matrix becomes rank one instead.
>
> We apologize for the confusion. In the revision, we have removed the phrases "for simplicity of presentation" and
> have moved the discussion about the case with general weights right after the proof of Theorem 2 at the bottom of page 7.
>
> **Q4: More details for Theorem 1.**
>
> We explicitly noted in Theorem 1 that the result holds for any level of noise in label noise SGD. We further reminded the definition of label noise SGD to the reader right before stating Theorem 1.
>
> We thank the reviewer for the valuable references provided; indeed obtaining non-asymptotic results for label noise SGD is more demanding and we see it as an important future direction, even for more general networks. We have added the provided references to the paper.
>
> **Q5: Typos.**
>
> Thank you for spotting the typos. We just want to point out that regarding your comment about page 3 typos there is probably a misunderstanding, the line “the first phase where the algorithm is far from stationary points trace of Hessian…” should be read as “the first phase where the algorithm is far from stationary points, trace of Hessian decreases slowly and...”. We have added the comma after “points” to make that clear.
>
> Given the changes we have made, we kindly ask the reviewer to increase her score.

---

### Official Review · Reviewer_GLiD · 2023-11-09

**Soundness:** 3 good
**Presentation:** 3 good
**Contribution:** 3 good
**Rating:** 6
**Confidence:** 3

**Summary:**

This paper studies the implicit bias of SGD on two-layer neural networks. The paper theoretically proves that:
1. A variant of SGD (with label noise) converges to the global minimizer with zero loss under a small learning rate;
2. The converged global minimizer has a globally minimal trace of Hessian among all global minimizers of the network;
3. At the converged point, for each data, the pre-activations are the same among all hidden neurons.
Note that 2 and 3 together imply that the flattest minimum (i.e., with the smallest Hessian trace) is also the "simplest" minimum in the sense of having a rank-one feature matrix. Thus, the simplicity bias and sharpness minimization bias suggest the same solution in this setting.

**Strengths:**

1. Using a very clean framework, this paper theoretically justifies two popular conjectures: 1) sharpness-reduction implicit bias implies simplicity bias in neural networks; 2) label noise SGD converges to the global minimizers of the sharpness on the manifold of zero loss.

2. This paper considers a much more general framework than existing work, i.e., a two-layer neural network (with arbitrary width and fixed output weights.

**Weaknesses:**

1. The assumption of fixed output weights avoids a key difficulty in analyzing neural networks --- symmetry in the output space. Such symmetry renders the global minimal to form a connected plateau. Also, excluding the output weights in optimization space also avoids dealing with variable coupling. This simplification unavoidably loses some important nature in neural network training.

2.  The assumption of a larger input dimension than the number of samples deviates a lot from practical settings. In particular, such an assumption (together with the choice of activation) ensures the realizability of the model (i.e., zero minimal loss) regardless of the network width. However, in most practical settings, the input dimension is much smaller than the number of data samples, and the realizability is usually achieved by sufficient network width.

**Questions:**

Other questions and concerns:

1. There are a number of typos. For example, on page 4: "Equation equation 1", "non=degeneracy", "equipp".

2. Page 4: "This is mainly for the ease of exposition and our results generalize to any fixed choice of weights of the second layer, in which the φ′′ image of the feature matrix becomes rank one." I don't understand why this is related to φ′′ (I suppose this is the second-order derivative of φ).

3. Page 5: "In Lemma 15, we show that θ(t) remains in a bounded domain along the gradient flow, which means having the weaker assumption that φ′, φ′′′ > 0 automatically implies Assumption 1 for some positive constants ρ1 and ρ2." I am not fully convinced by this claim. Lemma 15 is also established based on Assumption 1. Thus, if we do not have positive lower bounds on φ′, φ′′′, Lemma 15 may not hold and θ(t) may not stay in a bounded region.

---

> ### Author Response · Authors · 2023-11-20
> **Response to reviewer GLiD**
>
> **Q1: (typos)**
>
> We thank the reviewer for spotting the typos,  we have fixed them in the new version.
>
> **Q2: Case of unequal weights and connection between $\phi$ and $\phi''$**
>
> We only consider the case of equal weights, but in the general case with unequal weights, it turns out that with a similar analysis as in the proof of Theorem 2, the $\phi''$ image of the feature matrix becomes rank-one. We have moved the discussion about the general case which was within the proof of Theorem 2 to the end of Section 4.2. We apologize if this created any confusion.
>
> **Q3: The flow might not stay in a bounded region without Assumption 1**
>
> We thank the reviewer for pointing out to this typo in the statement of Lemma 15; indeed, Lemma 15 only requires the positivity of the third derivative $\phi''' > 0$, and does not depend on Assuumption 1. Therefore, our argument for obtaining Assumption 1 from a weaker positivity assumption on $\phi',\phi'''$ is not circular. We corrected this typo in the new version that we uploaded.
>
> Given our changes and feedback, we kindly ask the reviewer to increase her score.

---

> > ### Comment · Reviewer_GLiD · 2023-11-23
> > **Thank you for the response**
> >
> > Thank you for the response. After reading other reviewers' comments as well as the authors' responses, I decide to keep my score.

---

### Author Response · Authors · 2023-11-20
**Response to the reviews**

We thank the reviewers for their valuable feedback on our work. We have updated the paper based on this feedback. We have also uploaded our code.

Below, we have replied to each reviewer's comments separately.

---

### Meta-Review · Area_Chair_AttD · 2023-12-15

**Metareview:**

This paper analyses the sharpness aware minimization to train 2-layer neural networks. The authors showed that the label noise SGD actually minimizes the flatness and the flattest solution yields rank 1 solution in a high dimensional setting.

This paper gives an interesting theoretical implication. On the other hand, there are some issues. First, the quantitative convergence analysis in Theorem 1 is not well justified. As the reviewer xXHB pointed out, the existing paper (Li et al., 2021) actually did not give such a quantitative evaluation. Although the authors mention that it is not a problem, I also think this part is not rigorous. Otherwise, it should have additional justification. Another (rather minor) concern is that some conditions are a bit restrictive. For example, the activation function is not usually used one and the input dimension should be larger than the sample size.

In summary, I cannot recommend acceptance for the current form of the paper. I encourage the authors to proof read the paper again (to remove typos completely) and make the theorems more rigorous.

**Justification For Why Not Higher Score:**

As I mentioned in the meta-review, some part of the mathematical analysis is not rigorous. The existing work result is not correctly referred. Hence, I recommend rejection.

**Justification For Why Not Lower Score:**

N/A

---

### Decision · Program_Chairs · 2024-01-16

Reject